# Effectiveness of Ecosystem Strategies for the Sustainability of Marketplace Platform Ecosystems

**Yuki Inoue** [1,2,*] **, Masataka Hashimoto** [3] **and Takeshi Takenaka** [2]

1   Graduate School of Social Sciences, Hiroshima University, Higashi-senda, 1-1-89, Naka-ku,
    Hiroshima 730-0053, Japan
2   Human Augmentation Research Center, National Institute of Advanced Industrial Science and Technology,
    6-2-3, Kashiwanoha, Kashiwa, Chiba 277-0882, Japan; takenaka-t@aist.go.jp
3   Graduate School of Global Business, Meiji University, 1-1, Surugadai, Kanda, Chiyoda-ku, Tokyo 101-8301,
    Japan; m_hashi8@meiji.ac.jp
*   Correspondence: yuinoue@hiroshima-u.ac.jp

**Abstract:** Physical intermediary firms, such as logistics firms, are the foundation of marketplace platform ecosystems. This study introduces the case of a delivery crisis caused by the withdrawal of major logistics firms from the Japanese marketplace platform. To address such a problem, this study considers the application of an "ecosystem strategy". We define an ecosystem strategy in this situation, as "the strategy by which the platform owner cooperates with logistics firms to standardize logistics services and provides a platform system to improve cooperation among them". We constructed an agent-based simulation system customized by a dataset of Japanese platform-based markets to test the effectiveness of the proposed strategy. The results indicate that the introduction of the ecosystem strategy postponed the start of the collapse. It also increased the number of platform users by roughly 1.10 times and increased the total profits of logistics firms about 1.22 times. Additionally, it removed the trade-off relationship between platform users and the profits of logistics firms and allowed the maximization of both. This study contributed to the research stream of platform ecosystems by defining an ecosystem strategy, including physical intermediary firms, and verifying the effectiveness of the strategy for ecosystem evolution and sustainability.

**Keywords:** platform-based market; platform ecosystem; marketplace platform; e-commerce; delivery crisis; ecosystem strategy; service open innovation

## 1. Introduction

### 1.1. Marketplace Platforms and Physical Intermediary Firms

Marketplace platforms have developed extensively in recent years. In a marketplace platform model such as Amazon.com, third-party sellers (and the platform owner) provide the products to be sold on the platform, consumers buy the products via the platform website, and the platform arranges for their delivery to the consumer [1]. Researchers have considered platform owners, third-party complementors (product or service providers), and consumers as major actors [2–8]. However, our focus is on firms that are physical intermediaries of products. In this study, such firms are referred to as "physical intermediary firms". Physical intermediary firms are significant actors in platform-based markets where things are physically conveyed. For example, marketplace platforms (e.g., Amazon.com) need logistics firms such as physical intermediary firms. Physical intermediary firms are the foundation of marketplace platform ecosystems. Without them, the platform business model could not be implemented. Furthermore, the quality of service that physical intermediary

firms provide affects platform consumers. Accordingly, we deem physical intermediary firms to be as important as third-party complementors are to platform-based markets. Table 1 shows an example of physical intermediary firms on different types of platforms.

**Table 1.** Example of physical intermediary firms on different types of platforms.

| Platform Type | Example of Platforms | Complementors | Intermediator of Transactions | Intermediator of Goods | Interface with Consumers |
|---|---|---|---|---|---|
| Hardware platform | Nintendo Switch; PlayStation 4 | Software provider | **Retail shop** | **Logistics firm** | **Retail shop**; Interface of the platform |
| Application and/or software download platform | Google Play; App Store; PlayStation 4; | Software and/or application provider | Platform | - (Internet supplier) | Interface of the platform |
| Marketplace platform | Amazon.com | Product provider | Platform | **Logistics firm** | **Logistics firms**; Interface of the platform |
| Service intermediary platform | Expedia, Hotels.com | Service provider | Platform | **Transporter (for service customers)** | **Service provider**; Interface of the platform |

Note. Actors that are in bold and underlined refer to physical intermediary firms.

*1.2. Delivery Crisis in the Marketplace Platform Ecosystem*

We have confirmed a tendency for platform owners to neglect physical intermediary firms. As an example, we introduce the case of a "delivery crisis" that was caused by the withdrawal of a major logistics firms from the Japanese marketplace platform. Major firms in Japan (Yamato Transport Co., Ltd., Sagawa Express Co., Ltd., and Japan Post Holdings Co., Ltd.) have captured most of the home-delivery market and the e-commerce delivery business, including platform-based markets [9]. However, the large volume of Amazon transactions exceeded the capacity of one of the three major logistics firms, Sagawa, forcing it to subcontract logistics firms [10]. Amazon's high requirement for logistics firms and low prices nearly crippled Sagawa [10]. Finally, Sagawa ended the majority of its partnership with Amazon in 2013 [10]. The remaining two major logistics firms (Yamato and Japan Post) acquired these abandoned transactions [10]. However, the increasing volumes from Amazon worsened the working environment in Yamato [11]. Subsequently, Amazon (and its consumers) agreed to raise the delivery fees in 2016 [12]. However, ultimately, Yamato decided to withdraw a portion of its Amazon delivery business (same-day delivery service) to improve its working conditions [13]. Amazon then covered this shortfall using minor logistics firms [14]. However, as these minor firms could not provide the same quality of services as the major logistics firms, consumers began receiving lower-quality services from the platform [14]. We consider that Amazon's ecosystem became not to exert previous values because of lower service quality and higher delivery fees. However, in the worst case, if Amazon (or its consumers) had not accepted the increase in delivery fees, the ecosystem would have collapsed after the larger scale of withdrawal of major logistics firms.

In spite of the importance of these firms, we also consider researchers of platform fields have not focused on physical intermediary firms. We deem the reason might be because such works are considered to be the domain of researchers focused on logistics management. Researchers started to focus on matters concerning e-commerce delivery delays from about 1999. They studied the optimization of delivery specifically for e-commerce endeavors [15]. Researchers suggested that the traditional focus of such studies have been on the optimal placement of logistics centers and warehouses, stock and distribution in logistics networks, and algorithms of delivery and routing [16]. Some researchers focused on the management of delivery prices according to delivery situations. Ha, Li, and Ng [17] suggested that logistics firms acquire a sense of competitiveness due to the high frequency of deliveries and such situations increase price competitions among other logistics firms. Zhang, Gou, Yang, and Liang [18] investigated the pricing policies of e-commerce enterprises in situations of service quality declines during busy seasons and suggested that long-term service quality declines lead to lower product prices.

We deem that these findings from previous research can be applied to physical intermediary firms on marketplace platforms. However, the occurrence of the Japanese delivery crisis implies that mere logistics management would face limitations in their capacity to handle increasing demand for marketplace platforms. We believe that a new approach is needed with regard to ecosystem collapses that are caused by physical intermediary firms in platform-based markets.

### 1.3. Research Question and Purpose

As a new approach, this study considers the application of an "ecosystem strategy" for enabling physical intermediary firms to handle such problems. In the research field of management, researchers have recently developed an ecosystem strategy [19] (p. 47) that defines the ecosystem by the "alignment structure of the multilateral set of partners that needs to interact in order for a focal value proposition to materialize". Furthermore, it defines an ecosystem strategy as "the way in which a focal firm approaches the alignment of partners and secures its role in a competitive ecosystem". Applying this strategy to the relationship between the platform owner and physical intermediary firms, we consider that the platform owner approaches the alignment of physical intermediary firms. Although there are several ways to secure its role, this study considers the sustainability of the ecosystem as a platform owner by maintaining sustainable physical intermediary services. Therefore, the platform owner tries to enhance the capacity of physical intermediary firms by improving their alignment relationship. This can be another approach to the sustainability of a platform-based marketplace, instead of logistics management [16–18].

Here, current researches on platforms have also focused on ecosystem perspectives and developed the concept of "platform ecosystems" [1,7,20–22]. However, besides some studies (e.g., [1]), as far as we know, no study has included physical intermediary firms as significant actors in platform ecosystems. Therefore, this research gap provides our research question as follows:

*RQ: In marketplace platform ecosystems, is ecosystem strategy approaching physical intermediary firms effective for delivery crisis?*

We define ecosystem strategy in this study as "the strategy with which the platform owner approaches the alignment of physical intermediary firms to improve their capacity to maintain sustainability of the ecosystem". We will revise this definition with more precision for the hypothesis testing in the subsequent section (especially in Section 2.2.2). Thus, the aim of this study is to test how the application of an ecosystem strategy on physical intermediary firms can contribute to circumventing the collapse of platform ecosystems as well as facilitating the evolution and sustainability of the platform-based market. Since the phenomena of delivery crisis in our research question occurred in the Japanese marketplace platform, this study investigates the Japanese platform-based marketplace.

This study developed an agent-based simulation system of a platform ecosystem. We customized this system by a dataset related to Japanese marketplace platforms. From the simulation results, we succeeded in computationally confirming the situation of the Japanese delivery crisis and the future ecosystem collapse. Furthermore, we confirmed that the application of an ecosystem strategy for physical intermediary firms is effective both in avoiding the collapse of platform ecosystems and in improving the profits of actors within the ecosystem. Specifically, the platform owner could avoid such a collapse with a smaller increase in additional delivery fees if the ecosystem strategy was applied. The application of this strategy would also provide about 1.10 times the number of platform transactions and about 1.22 times the profits of logistics firms.

The remainder of our paper is organized as follows. In Section 2, we review the related literature and develop a research hypothesis. Additionally, we describe our methods, including the structure of an agent-based simulation system and an explanation of the data analysis. Some detailed parts of the methods are described in the Appendix A. In Section 3, we show the simulation results. In Section 4, we discuss the implications of our study and make suggestions for future research. Finally, in Section 5, we conclude this study.

## 2. Materials and Methods

*2.1. Literature Review*

In this subsection, we review the related literature about platform ecosystems, its evolutional mechanisms, and the methodology of agent-based simulation.

### 2.1.1. Platform Ecosystems

A recent major topic in platform research concerns platform ecosystems [20,21]. A platform ecosystem is made up of systems or architectures that are supported by a collection of complementary assets [21,23,24]. Among complementary asset providers, producers of complementary goods for the platform are called the complementors [25]. A platform ecosystem can foster unlimited innovation via the participation of various organizations that hold several management resources as complementors [20]. It also leads consumers with various needs to adopt the platform [26].

The platform ecosystem has been extensively researched and various types of platforms have been considered. For example, the video game market is a platform ecosystem. It consists of hardware as the platform, software as a complementary good, software providers as complementors, and the consumers who purchase the final products [2–8]. The focus of research on platform ecosystems varies among the evolution of platform ecosystems and competition (and co-evolution) among platform ecosystems [2,3, 22,27,28], the growth mechanisms of complementors [26,29], the diversity of complementary goods provided by complementors [30], the development of competition within a platform ecosystem [31], intergenerational platform-technology transitions [32], integration among platform ecosystems [33], intellectual property and the technology of platforms and complementors [34–38], governance of ecosystems [39], heterogeneity in platform-based markets [40], and sustainability of ecosystems [41]. This study can be classified as a research stream of the evolution of platform ecosystems. We contribute to the research stream by studying the perspective of physical intermediary firms.

### 2.1.2. Evolutional Mechanisms of Platform Ecosystems

One of the most significant mechanisms of platform ecosystems is the "indirect network effect" [42]. The indirect network effect signifies that in a two-sided market, as scale grows on one side, profit increases on the other [43–46]. As this effect implies that complementors and consumers interact with one other and the number of complementors and customers grow exponentially, the result could be a winner-takes-all market where a single platform absorbs almost all complementors and consumers [47–49].

Establishing an installed base is important in generating indirect network effects [2,3,50]. A platform with a smaller installed base with no specialized markets would face negative growth due to the indirect network effect [48]. In addition, as superior complementary goods promote the total sales of the platform [51], platform owners often offer attractive complementary goods themselves [5,52] or encourage capable complementors to do so [53]. However, even if a platform achieves a large installed base, lower volumes of complementary goods would cause its ecosystem to decline [54]. Accordingly, the platform owners should pay attention not only to the profits of its consumers but also to the profit of the complementors.

Therefore, researchers have suggested that the evolution of the platform ecosystem is driven by the establishment of installed bases and the variety of complementary goods to provoke indirect network effects. However, in the case of general product intermediary platforms, these natures might be somewhat different. Consumers and third-party sellers can generally register on multiple marketplace platforms. Unlike the case of video games or other hardware platforms, consumers on marketplace platforms do not incur many expenses to start with; they can easily register themselves on the platform website and access it. Some marketplace platforms do provide premium services (e.g., Amazon Prime). However, these services are for loyal customers of the platform. Additionally, as each platform can easily be accessed through internet searches, there is almost no physical barrier to using

multiple platforms. In the case of third-party sellers, the initial cost is low and typically only consists of a platform usage fee. Therefore, platform owners must constantly offer benefits to consumers and third-party sellers to maintain or increase the number of platform participants. An effective way of doing this is by introducing a pricing scheme for usage fees and incentive settings. Therefore, we will review previous studies of pricing on platform-based markets.

Studies on platform price-setting methods have been conducted mainly within the context of two-sided markets. An early definition of a two-sided market (or multi-sided market) was "markets in which one or several platforms enable interactions between end-users and try to get the two (or multiple) sides 'on board' by appropriately charging each side" [45] (p. 645). Currently, the scope of the two-sided market (or multi-sided market) has broadened to not only include end-users but also various organizations [46]. As a price-setting strategy, Yoo, Choudhary, and Mukhopadhyay suggest that higher platform fees should be imposed on those who can enjoy a more effective indirect network effect [55]. Rochet and Tirole state that platform price allocation should be based on demand elasticity in accordance with the Ramsey pricing method [44]. Caillaud and Jullien suggest that the best strategy to dominate the market and protect market shares entails setting low participation costs (possibly subsidized) and maximum possible transaction fees [56]. Armstrong [57] and Rochet and Tirole [44] focus on single-homing and multihoming among multiple platforms. They indicate that when one side is single-homing while the other is multihoming, a high price should be charged on the multihoming side. In the next subsection, we consider the application of such a pricing scheme on the relationship between consumers and physical intermediary firms.

### 2.1.3. Agent-Based Simulation

This study reproduces the transactions of the marketplace platform ecosystem in Japanese markets using agent-based simulation. This method simulates the behavior of actors (agents) that comprise the social system, especially how they act to influence others [58]. The system is programmed on a computer, and the agent attempts to reproduce the real exchanges among the actors by autonomously making decisions and interacting in the artificial environment [59].

Although the agent-based simulation approach is not a commonly used technique in strategy and management, some studies have applied it to study the impact of new enterprises on the market environment [60], spread of innovation [61], impact of consumer purchasing behavior on other consumers [62], effectiveness of management strategies of retail chain stores [63], cooperative network formation in business ecosystems [64], and competition between platforms [65]. In addition, some studies examine platform ecosystems using agent-based simulation methods [41]. Thus, studies have used the agent-based simulation approach to examine the interactions between agents. The platform ecosystem in this study also consists of interactions between agents, and the outcome of the whole ecosystem can change because of emergent phenomena caused by their interactions. Therefore, the application of the agent-based simulation approach is appropriate for this study.

### 2.2. Conceptualization and Hypothesis

In this subsection, we consider the mechanisms behind the collapse of marketplace platform ecosystems. Then, we develop a research hypothesis related to the application of an ecosystem strategy on physical intermediary firms.

### 2.2.1. Mechanisms behind the Collapse of Marketplace Platform Ecosystems

As described in Section 2.1.2, previous studies focused on pricing in terms of usage fees between the two sides. We believe this approach can be applied to consumers and third-party sellers. However, we also consider the difficulty in applying this theory to the relationship between consumers and physical intermediary firms, especially logistics firms. As the profits for consumers decrease as platform usage fees increase, the demand elasticity of consumers is relatively large. Meanwhile, if logistics firms cannot acquire delivery orders, they have to keep employees and transportation vehicles as idle

assets. Accordingly, the demand elasticity of logistics firms is relatively smaller than that of consumers. Conversely, logistics firms can benefit from the indirect network effect as their profits increase as the number of consumers increases. Besides, from the perspective of the marketplace platform, there is no difference in terms of the type of homing between consumers and logistics firms; both consumers and logistics firms are basically multihoming. Therefore, we suppose that consumers have the upper hand in determining fees over logistics firms. This situation would not be problematic until the delivery amount reaches the upper limit of the logistics firms' capacities. This is because the difference between the upper delivery limit and the delivery amount becomes dead stock, as is the nature of the logistics business.

However, after the delivery amount exceeds the upper capacity limit of the logistics firms, they must force their employees to work overtime or devise a plan to outsource the deliveries. Thus, we deem that the profit functions of logistics firms would be changed before and after the excess. If we simply formulate such a change of profit functions and set $y$ as the profit of logistics firms, $x$ as the number of deliveries, and $a$ as the delivery fees, we can define the function of the pre-excess situation as $y = ax$. If we set the upper delivery limit of as $L$ and the additional overtime work fee or additional outsourcing fee per delivery unit as $b$, we can define the function of the post-excess situation (i.e., $x > L$) as $y = aL - b(x - L)$. Ultimately, when the profit of the logistics firm becomes zero or becomes lower than the expected profits of other businesses, the logistics firm would withdraw from the platform. If any major logistics firm withdraws from the platform, the delivery amounts of the remaining logistics firms would, in turn, drastically increase. At this moment, if the platform owner does not support the increases faced by the remaining logistics firms (e.g., by increasing the delivery fee), they would also withdraw from the platform. Finally, since the amount of handled product deliveries becomes fewer, the platform ecosystem would become unable to function as a marketplace. We define this situation as the "collapse of an ecosystem". Additionally, we define the boundary consisting of the delivery demand and logistics capacity, which resulted in the collapse of the ecosystem, as the "boundary of collapse". If we do not consider the growth of delivery capacity in time, we can observe the boundary of collapse as the combination of the amounts of platform transactions and delivery fees on the marketplace platform. Figure 1 shows the overview of a collapse of an ecosystem and the boundary of collapse. Additionally, Figure 2 shows the process leading to the collapse of ecosystems. We consider the case of the Japanese delivery crisis as having followed the sequence of process 4b (at least in the current situation) as in Figure 2.

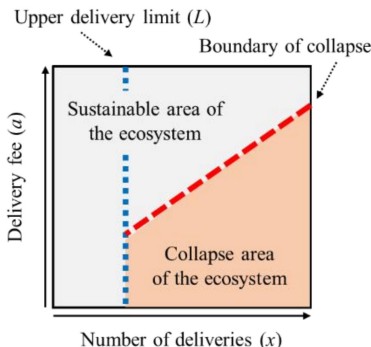

**Figure 1.** Overview of the ecosystem collapse and the boundary of collapse. The boundary is not limited to only being linear.

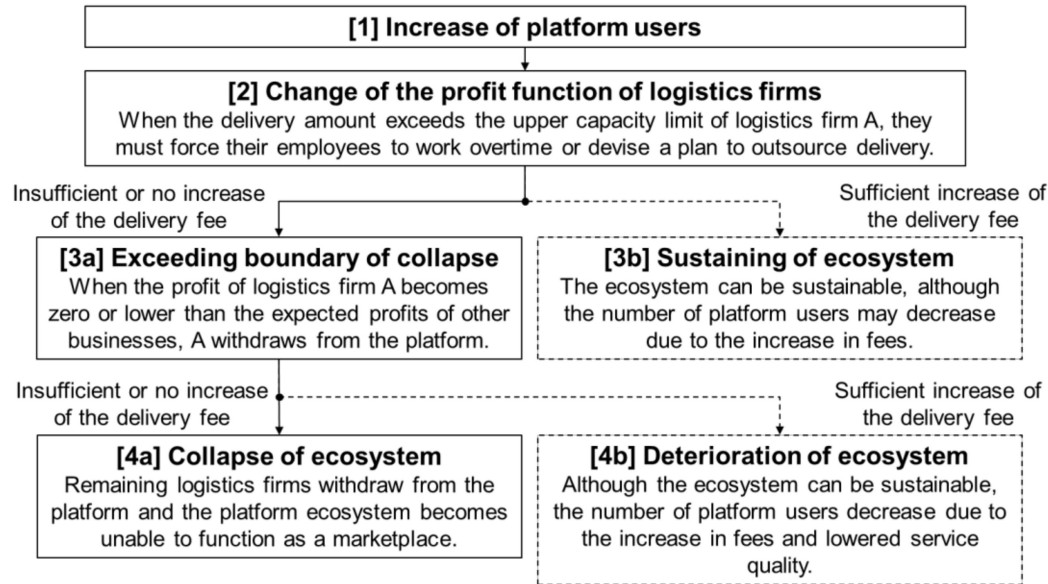

**Figure 2.** Process reaching collapse of ecosystems. If the delivery capacity of logistics firm A is sufficiently small, it may not reach process 4 after the passage of process 3a. However, in the case of the Japanese delivery crisis, the withdrawal of logistics firm A (Sagawa) sufficiently influenced the remaining logistics firms.

### 2.2.2. Application of the Ecosystem Strategy for Physical Intermediary Firms

In this section, we consider how the application of an ecosystem strategy for physical intermediary firms on marketplace platforms influences the prevention of the ecosystem's collapse. Adner [19] (p. 47) defines an ecosystem strategy as "the way in which a focal firm approaches the alignment of partners and secures its role in a competitive ecosystem". Additionally, this study defined the ecosystem strategy in this study as "the strategy which the platform owner approaches alignment of physical intermediary firms to improve capacity of them for keeping sustainability of the ecosystem". Since our focus is on platform-based markets, we consider the application of an ecosystem strategy in terms of the concept of platform ecosystems.

In a recent study, Jacobides, Cennamo, and Gawer [66] suggested a way of realizing platform ecosystems. They classified types of complementarities in terms of production and consumption as generic, unique, and supermodular [66]. A detailed definition of each term is included in the paper, but we will simply summarize them. Generic complementarities refer to the "production or consumption of items can be independently each other", unique complementarities mean that the "joint production or joint consumption of items is either mandatory or has superiority over their independent production or independent consumption", and supermodular complementarities mean "more production or more consumption can bring benefits to other items". Additionally, they predicted that when both production and consumption are higher than or equal to unique complementarities, they can be regarded as an ecosystem.

If we consider the application of these definitions on physical intermediary firms and consider the production and consumption of delivery services, we can find room for more ecosystemization. As an element of production, we confirmed that logistics firms provide specific delivery services for the marketplace platform in Japan. Specifically, the Amazon platform provides an Expedited Shipping service as part of Amazon Prime and the logistics firms handle this specialized delivery service. One of the logistics firms, namely Yamato Transport Co., Ltd., a major actor in Japan, developed its e-commerce service systems (e.g., Today Shopping Service or TSS and Free Rack Auto Pick System or FRAPS) to facilitate close cooperation between Amazon and the delivery centers of Yamato. Therefore, we consider the platform owner and the logistics firm to jointly provide a unique delivery service. Additionally, since the created service must be achieved by joint use between the platforms' special

delivery plans and the specialized logistics firms, the consumption side is also regarded as unique. Based on these considerations and the concepts of Jacobides, Cennamo, and Gawer [66], we can derive an ecosystem strategy wherein the platform owner provides its platform systems and policy to facilitate coordination among logistics firms that improve the complementarity level of production and consumption from unique to supermodular.

As an example, we propose the following coordinated delivery scheme. First, the platform owner makes logistics firms declare (or measure in some way) its delivery capacity and provides a delivery allocation system to equally distribute the delivery orders based on that capacity. Second, the platform owner coordinates with logistics firms and develop a system that allows each logistics firm for the co-use of the delivery base with other logistics firms. The realization of this scheme would be achieved by standardization (or modularization) of these systems by the platform owners and the adaption of the standard (or module) by logistics firms. As logistics firms join these systems, the total delivery efficiency increases, and the total delivery capacity becomes larger. This means that the average capacity of the delivery service increases and the increased service quality (caused by preventing each firm's capacity to be exceeded) can, in turn, increase the consumers' profit. Therefore, we deem that this scheme could improve the complementarity level of production and consumption from unique to supermodular. Additionally, since this scheme increases the total capacity of logistics firms, the boundary of collapse may move towards the right in Figure 1 and decrease the risk of the ecosystem collapsing. Since this means lower delivery fees are needed to sustain the platform-based market in large platform transactions, we predict that a larger platform user base and larger logistics profit may be realized. Thus, we propose the following hypothesis:

*The introduction of an ecosystem strategy, which facilitate standardization and cooperation among physical intermediary firms, can (a) decrease the risk of an ecosystem collapse and (b) improve the evolution of the platform ecosystem by increasing the number of platform users and the profits of physical intermediary firms.*

In summary, we finally defined the ecosystem strategy in this study as "the strategy by which the platform owner cooperates with logistics firms to standardize logistics services and improves platform system cooperation among them". In the simulation, this strategy has two effects as follows: (a) the platform owner makes logistics firms declare their delivery capacity and provides a delivery allocation system to equally distribute the delivery orders based on that capacity, and (b) the platform owner provides a system that facilitates each logistics firm to co-use the delivery base with other logistics firms.

### 2.3. Analysis Framework

This study tests how the introduction of an ecosystem strategy can influence marketplace platform ecosystems. However, we cannot conduct an empirical approach to the statistical analysis since there are no existing cases as far as we are aware. Therefore, this study takes the approach of an agent-based simulation to test our hypothesis.

Figure 3 shows the structure of the platform ecosystem assumed in this study. We referred to the structure of platform ecosystems expressed by previous studies (e.g., [7,41]) and revised them for marketplace platforms, including logistics firms. Consumers purchase the products of third-party sellers via the platform. The third-party sellers deliver the products to consumers through logistics firms. The platform can receive usage fees from third-party sellers and provide discounts as an incentive for consumers. Logistics firms receive delivery fees from the platform. For simplicity, our simulation does not include the stock of products of platform owners.

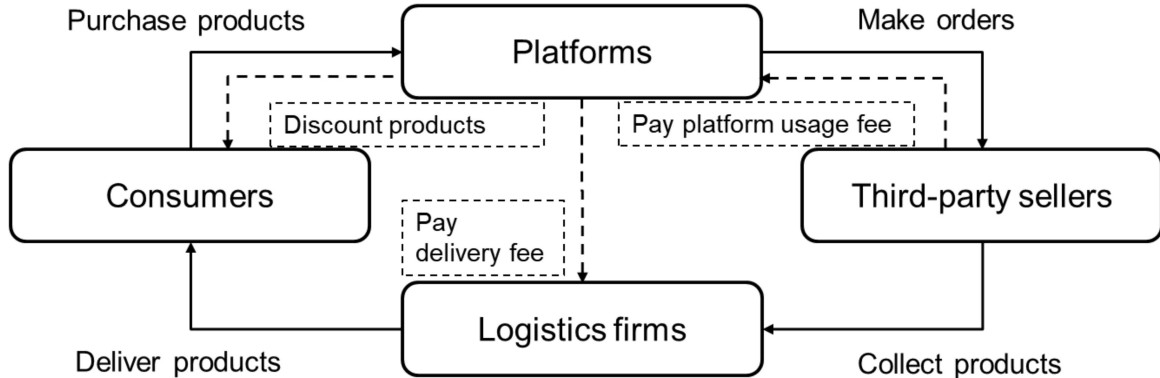

**Figure 3.** Structure of the platform ecosystem assumed in this study.

Some portion of researchers using agent-based simulations may argue that the method should be simplified as far as possible and should not use a real dataset. However, we consider such an approach to be inappropriate for an analysis of platform ecosystems, since they include interactions among various types of actors and generate problems of change in results depending on the parameter settings of the agents. Additionally, some researchers highlighted the significance of empirical data in agent-based simulation. Boero and Squazzoni [67] suggested that empirical knowledge needs to be appropriately embedded into modeling practices through specific strategies and methods. They also argued that empirical data are needed to build sound micro specifications of the model and to validate the macro results of the simulation. Furthermore, models should be both empirically calibrated and empirically validated. Therefore, this study uses data related to the Japanese platform-based marketplaces and specialized the simulation system according to the Japanese market.

### 2.4. Structure of the Agent Simulation

For simplicity's sake, our simulation mainly focuses on the interactions between consumers and logistics firms. Therefore, our simulation does not include more detailed information about sellers, such as the participating mechanisms of individual sellers and features of the products. Figure 4 shows the simulation process. Our agent-based simulation included four types of agents, namely the platform agents, consumer agents, logistics firm agents, and third-party seller agents.

The simulation procedure is as follows: (a) The number of consumer agents which can use platform is updated. (b) Each consumer agent decides to purchase a product from either a platform or from other sources. The consumer decides by considering various factors, including the price discount on the platform, the product variety, the probability of delay in delivery, and the delivery period as part of the delivery service. (c) The number of platform users at the simulation step is updated. The platform allocates the delivery of goods purchased on the platform to each logistics firm agent. The third-party seller agents change the scale of provision of their products on the platform according to the scale of the consumers' platform usage as an indirect network effect. (d) The logistics firm agents deliver as many products as possible at their delivery capacity to each consumer's home from their own delivery bases. This simulation deals with the so-called "last mile" from the delivery base to each consumer's home. We do not focus on the processes that take place before the last mile, namely the collection and shipping processes. They accumulate products at the delivery base nearest to the address of each consumer and deliver them to each house. If a consumer is absent at the time of delivery, logistics firm agents carry it back to the delivery base to re-deliver later. (e) For each certain period of simulation steps (we set 30 steps), the logistics firm agents change the degree of outsourcing as and when it is appropriate to do so. For simplicity's sake, our simulation considers outsourcing but does not consider the influence of employee's overtime work that comes as a result of excessive demand. (f) The logistics firm agents refer their past average profits for a certain period (we set 90 steps) and decide whether or not to withdraw from the platform when they continuously incur losses

of profits caused by excessive demand and insufficient delivery fees. (g) The value of the product variety on the platform and the value of the parameters of delivery services are updated. (h) If the simulation reaches a certain step, or if all major logistics firms agents withdraw from the platform, the simulation is finished. If not, the procedure returns to (a).

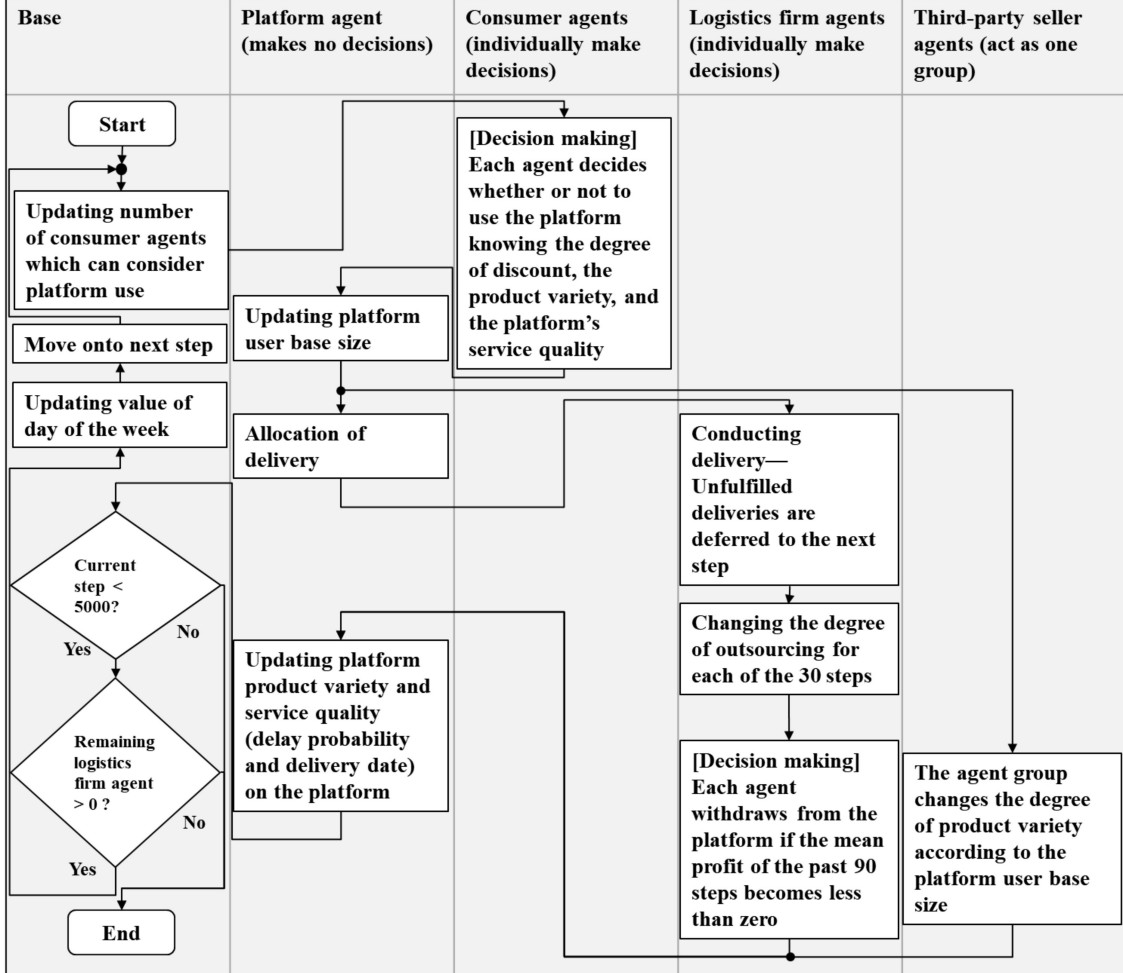

**Figure 4.** Simulation procedure.

This study used a lot of data to establish the parameter settings of agents, especially for consumer agents and logistics firm agents. For the consumer agent, we used public information on the distribution of location and age, current platform user base sizes, and the number of internet shopping users who are potential platform users. Additionally, we conducted a consumer questionnaire survey for marketplace platform users to build consumer decision models of platform use and set parameters on the use of platforms. For the logistics firm agents, we used public information on the home delivery shares of each logistics firm at a certain period. Additionally, we collected data from each logistics firm to acquire the number of delivery vehicles they possessed and establish the locations of the delivery base for last mile delivery from available information for each logistics firm. A more detailed explanation of these is shown in subsequent subsections and in the Appendix A.

### 2.4.1. Platform Agent

Our simulation included single-platform agents and did not consider competitive platforms. The platform agent acts as an intermediary between consumer agents and third-party seller agents and arranges the delivery for them. As per the e-commerce platform business model, the platform owner can reduce wholesale and retail fees, unlike the business models of retail stores. For simplicity's

sake, our simulation assumed that the revenue of the platform agent and the basic shipping fees for the logistics firm agents are paid for by the reduction of wholesale and retail fees. Thereafter, the simulation assumed that the platform usage fee (discount rate) of the consumer ($q^{pay}$) is paid for by the platform usage fee of third-party sellers. If consumers are charged an additional delivery fee $s^{logi.}$ by the platform in the simulation, the fee is deducted from $q^{pay}$. We refer the platform usage fee for third-party sellers of Amazon.com and set it to 0.1, i.e., 10% of the product price. If we suppose the standard value of $q^{pay}$ is about 0.1, the relation between $q^{pay}$ and $s^{logi.}$ is defined as

$$q^{pay} = -0.1 + s^{logi.}. \tag{1}$$

In actual cases, the platform owner can change the price settings between $q^{pay}$ and $s^{logi.}$ over time. However, we did not consider the time variations of the price setting and we set values for experimental conditions in the simulation because the effect of such changes of the price depends on the method used. Additionally, to simplify our research structure, our simulation does not test for price settings including third-party sellers. We reserve this for future research.

### 2.4.2. Consumer Agents

Our simulation included consumer agent $i$ of $N^{consumer}$ (i.e., $i \in N^{consumer}$). To reduce the calculation time of the simulation, as representative of the Japanese marketplace platform, one consumer agent in the simulation corresponded to 10,000 consumers in the market. Given that $N^{max.con.}$ is defined as the potential maximum size of consumers targeted by the platform, this simulation supposed that $N^{consumer}$ increases to reach the value of $N^{max.con.}$ through the simulation steps as the platform usage spreads to consumer agents. As reference values, the mean daily increase in the number of Japanese users of Amazon.com was determined to be 6380, based on the fact that Amazon.com was launched in November 2000 in Japan and had about 40 million users as of June 2018 [68]. However, when we applied this value (0.638 consumer agent increases per simulation step) in our simulation, the calculation time turned out to be too high. Therefore, we set $t$ as the unit of simulation steps, considering the days of the week to calculate $N^{consumer}$ as $6.38t$.

The consumer agents had the following parameters: probability of considering platform use, delivery appointment date and time, and inherent delivery acceptance dates and times. They also had inherent parameters for their influence on the platform usage fee (discount fee), namely the product variety on the platform and the delivery service quality, including delivery delay and delivery period. They decided on whether or not to use the platform based on these parameters and the situation that the platform was in. These parameters were based on the questionnaire survey and an analysis of the obtained survey results. They also had position parameters of home locations based on the demographic data. The consumer agent settings are described in further detail in the Appendix A.

### 2.4.3. Logistics Firm Agents

Our simulation includes four logistics firm agents $x$: the three major firms of Yamato Transport Co., Ltd.; Sagawa Express Co., Ltd.; and Japan Post Holdings Co., Ltd. (JP), and one agent intended for Others. We defined the allocation of the delivery ratios in each of these firms as Yamato: 0.46, Sagawa: 0.34, JP: 0.12, Others: 0.08. We assigned the ratio of home deliveries to the year 2013, which was when the delivery crisis started [69].

Each logistics firm agent $x$ has its own delivery bases and transport vehicles. We confirmed that Yamato has 3688 bases [70] Sagawa has 445 bases [71], and JP has 1088 bases [72]. We collected the address data of the delivery bases and converted them to latitudinal and longitudinal values. For the sake of simplicity, this study did not define the delivery bases of Others. We also confirmed that Yamato has 43,754 vehicles [70], Sagawa has 25,153 vehicles [71], and JP has 33,083 vehicles [72]. We assumed the number of vehicles of Others to be 8869, based on the delivery ratio (43,754 + 25,153 + 33,083) × 0.08. Here, these values refer to the sum of all transport vehicles. Based on pre-interviews held

with Yamato employees, the maximum delivery vehicles $g_x$ of agent $x$ were defined as 85% of the transport vehicles of agent $x$. The logistics firm agents deliver products to each consumer from their nearest delivery base. As our simulation regarded one consumer agent as representative of 10,000 actual consumers, the logistics firm agent implements 10,000 times more deliveries for each order.

Each logistics agent has delivery bases for the last mile, which have position parameters based on real data. They deliver goods from the delivery bases to each consumers' home. If the demand they receive exceeds their delivery capacity, they outsource the work to fulfill the delivery demands in exchange for losing their profits. They observe their own profits on the platform. If their profits from a certain period become less than zero, they decide to withdraw from the platform. These settings for logistics firm agents are described in further detail in the Appendix A.

### 2.4.4. Third-Party Seller Agents

For the sake of simplicity, since this study focuses on consumers and logistics firms, our simulation did not include detailed information about third-party sellers. Therefore, although we understand that actual third-party sellers make complex decisions on their platform usage (e.g., [73]), our simulation did not consider such parameters as the participating mechanisms of individual sellers and the features of products. Additionally, our simulation set third-party seller agents as an entire group of third-party sellers, that is, they do not act as individual agents. Third-party seller agents decide to change the degree of provision of their goods on the platform based on the size of the platform user base on the consumer side. The third-party seller agent settings are described in further detail in the Appendix A.

### 2.5. Simulation Experiments

### 2.5.1. Experimental Conditions

In our experiments, we tested cases where the proposed ecosystem strategy was not introduced and cases where it was and compared them. We tested the coordinated delivery scheme as an ecosystem strategy described in Section 2.2.2. The expressions in the simulation were as follows. (a) As shown in Section 2.4.3, we defined the delivery ratio allocation of each firm as Yamato: 0.46, Sagawa: 0.34, JP: 0.12, and Others: 0.08. When the proposed ecosystem strategy was introduced, the ratio was redefined as Yamato: 0.39, Sagawa: 0.23, JP: 0.30, and Others: 0.08 based on the ratio of the delivery capacity of each logistics firm by the number of delivery vehicles. (b) When the proposed ecosystem strategy was introduced, each logistics firm could use the delivery bases of other logistics firms for last mile delivery.

Within such tests, this study comprehensively conducted experiments in terms of the number of platform users and delivery fees parameters. The collapse of the marketplace platform ecosystems could be caused by exceeding the boundary of collapse of the logistics firms. This boundary is determined by the balance between consumer demands and the delivery capacities of the logistics firms. Therefore, this study conducted simulation experiments by changing the maximum number of consumer agents $N^{max.con.}$ and the additional delivery fees $s^{logi.}$. We confirmed that the number of current internet shopping users in Japan is at least 70 million. The value is acquired the total number of 95 million internet users in Japan multiplied by the rate of internet shopping users (72 percent). We obtained the number of internet users in Japan from e-Stat [74]. We also obtained the rate of internet shopping users from the White Paper on Information and Communication, 2015, of the Japanese Ministry of Internal Affairs and Communication [75]. We regarded this value (70 million) as the potential number of platform users. As mentioned previously, one consumer agent corresponds to 10,000 consumers and so we set the maximum upper limit of consumer agents in the simulation experiment at 7000. Additionally, we confirmed that the collapse of ecosystems would not occur until 3000 consumer agents were reached. Therefore, we set the minimum upper limit of consumer agents in the simulation experiment at 3000. We experimented with 9 patterns of $N^{max.con.}$ in increments of 500 within the range 3000–7000. We also experimented with 21 patterns of $s^{logi.}$ in increments of

0.005 within the range 0–0.1 as in Equation (1). Therefore, we conducted experiments based on the conditions of a total of 189 patterns.

While the agent parameters were refreshed when the $N^{max.con.}$ values were changed, they were not refreshed when the $s^{logi.}$ values were changed. This was because we specified the value of $s^{logi.}$ which corresponded with boundary of collapse or maximized the platform user base and the logistics firms' profits. The detailed analysis method is described in a subsubsection of the section entitled "Analysis of Simulation Results and Evaluation Indicators".

As we confirmed in advance that 4000 steps are enough for convergence, the simulation ran until it reached 5000 steps. Note that this convergence includes not only a convergence to a specific value but also a convergence to the limit cycles. We simulated each experimental condition 50 times and calculated the values of the consumer agents' platform usage rate as the platform ecosystem development index. Table 2 summarizes these parameter settings as experimental conditions.

**Table 2.** Parameter settings as experimental conditions.

| Parameter | Value |
|---|---|
| Introduction of the proposed ecosystem strategy | Not introduced or introduced |
| Additional delivery fee $s^{logi.}$ | 0, 0.005, 0.01, … , 0.095, 0.1 (21 patterns) |
| Maximum number of consumer agents $N^{max.con.}$ | 3000, 3500, 4000, … , 7000 (9 patterns) |
| Simulation steps $t$ | 1, 2, … , 5000 steps |
| Number of repetitions of experimental trial | 50 times |

### 2.5.2. Analysis of Simulation Results and Evaluation Indicators

Corresponding to the hypothesis, this study conducted an analysis from the following two perspectives.

First, we identified the boundary of collapse and compared it before and after the introduction of the ecosystem strategy. The procedure was as follows. (a) At any $N^{max.con.}$, and at any experimental trial, we acquired the number of remaining major logistics firm agents for each value of the additional delivery fee $s^{logi.}$ at the last step of the simulation. (b) The maximum value of the additional delivery fee $s^{logi.}$, which satisfied the situation when all major logistics firm agents withdraw, was defined as the boundary point of collapse at $N^{max.con.}$. (c) We collected the boundary point of collapse for all $N^{max.con.}$ and formed the boundary of collapse trajectory. The concept image was shown in Figure 1. (d) We confirmed the difference in the boundary of collapse trajectory before and after the introduction of the ecosystem strategy.

Second, we evaluated the degree of platform users and profits of logistics firms and compared them before and after the introduction of the ecosystem strategy. The calculation procedure was as follows. (a) For each simulation condition, we calculated the mean value of platform users (indicator A) and the mean total profit of three major logistics firm agents (indicator B) over the last 500 steps (from step 4501 to step 5000). (b) At any $N^{max.con.}$, and at any experimental trial, we respectively identified the values of $s^{logi.}$, which satisfies the maximum values of indicators A and B. (c) We collected the values of $s^{logi.}$ and maximizing indicators A and B for all $N^{max.con.}$ and formed their trajectory. (d) We confirmed the values of indicators A and B by following the trajectories made at (c) and comparing them to before and after the introduction of the ecosystem strategy, respectively.

## 3. Results

In this section, we show the simulation results and the correspondence between the results and the hypothesis. In the first subsection, we show the reproduction results of the Japanese platform-based marketplace and its collapse. In the second subsection, we show a comparison between the results before and after the introduction of the ecosystem strategy for physical intermediary firms.

### 3.1. Reproduction of the Japanese Platform-Based Marketplace

Figure 5 shows an example of the results of one experimental condition, where the maximum number of consumer agents $N^{max.cons.}$ is 4000 and the additional delivery fee $s^{logi.}$ is zero. Here, we show the changes in the number of consumer agents $N^{consumer}$ as inputs and the platform usage rate of consumers, the remaining number of major logistics firm agents, and the minimum delivery steps (logarithm) as outputs. The results demonstrate the process of the ecosystem's collapse as follows. First, the evolution of the platform ecosystem was smoothly achieved. Second, at step 720, a logistics firm agent decided to withdraw from the platform. The delivery environment subsequently became unstable. Delivery delays and an extension of delivery times occurred due to the increasing influence of a low-quality logistics firms. Additionally, these deteriorations of service quality decreased the number of platform users. Third, at step 990, the second logistics firm agent decided to withdraw from the platform. Thereafter, the deterioration of service quality worsened, further decreasing the number of platform users. Finally, at step 1890, the third logistics firm agent decided to withdraw from the platform. Ultimately, the platform ecosystem collapsed. Therefore, this result demonstrates the process that platform-based marketplaces undergo before the ecosystem collapses.

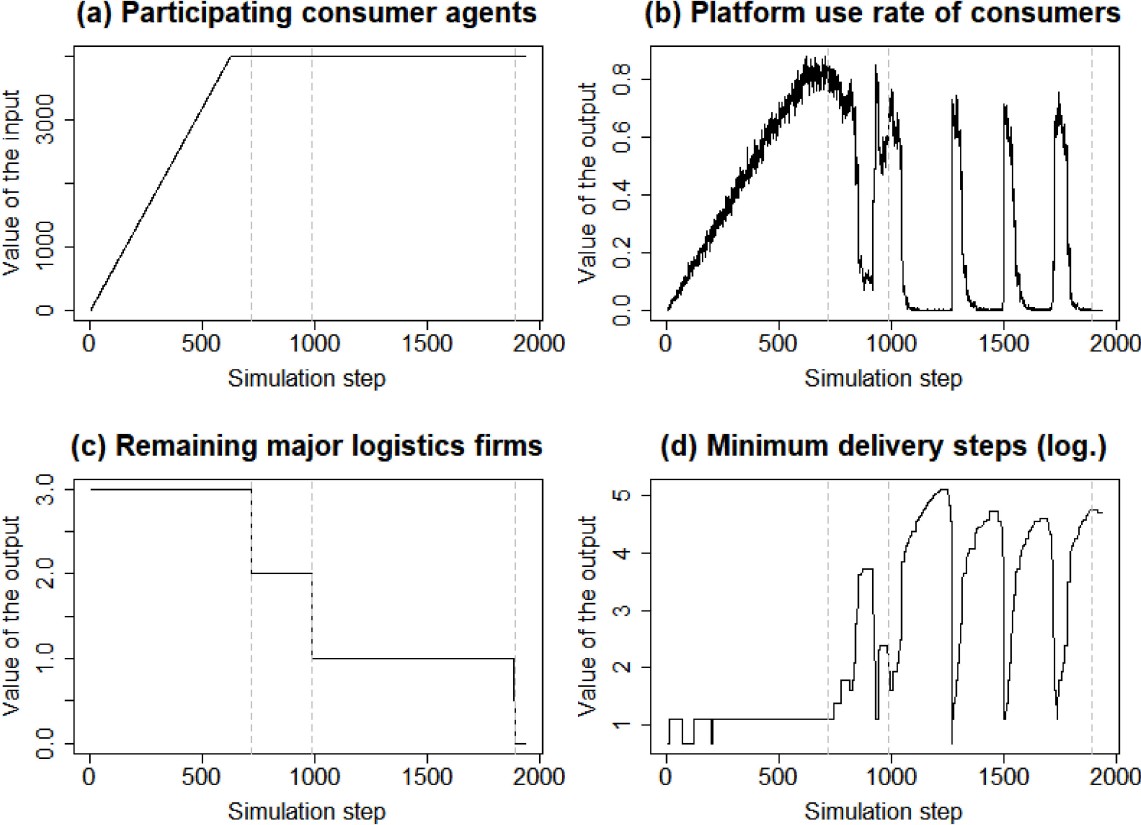

**Figure 5.** Example of the simulation results of one experimental condition. The condition is that where $N^{max.cons.}$ is 4000, the additional delivery fee $s^{logi.}$ is zero. Values of (**a**) are input in the simulation experiment. Values of (**b**–**d**) are acquired output.

Figure 6 shows the simulation results of all the combinations of $N^{max.con.}$ and any $s^{logi.}$. In each figure, the x-axis represents the $N^{max.con.}$ value and the y-axis represents the $s^{logi.}$ value. First, Figure 6a shows the change in the number of platform users by $N^{max.con.}$ and $s^{logi.}$. As the results indicate, when $N^{max.con.}$ was 3000, the number of platform users was at its maximum value with no additional delivery fee. However, when $N^{max.con.}$ was larger than 3000, the ecosystem collapsed. Therefore, to acquire the maximum number of platform users, an additional delivery fee was needed. Figure 6b shows the change in total profits of logistics firms. Although the tendency of the changes in values was

similar to that in Figure 6a, the trajectory maximizing logistics firms' profits was different to that of the platform users.

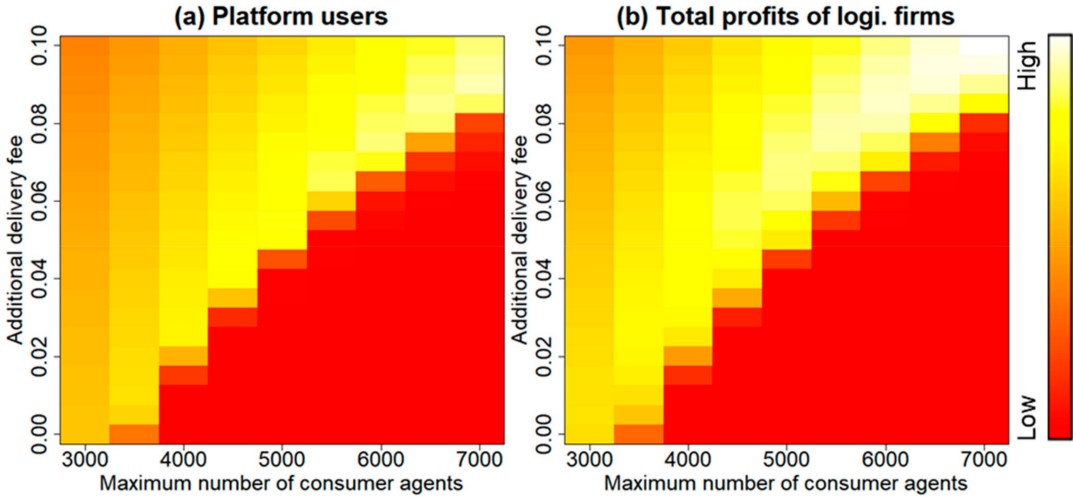

**Figure 6.** Simulation results of all combinations of $N^{max.con.}$ and any $s^{logi.}$.

Figure 7 shows the trajectory of the boundary of collapse and the delivery fees that maximize each indicator. Figure 7a shows the boundary of collapse. The results show that as $N^{max.con.}$ increases, the platform owners must charge consumers larger additional delivery fees. The results also indicate that the collapse of the ecosystem occurred drastically and the difference between the $s^{logi.}$ of the maximum value of platform users and the $s^{logi.}$ of the boundary of collapse was only about 0.005–0.010. Figure 7b shows the comparison between the trajectories of delivery fees that maximize each indicator. Until $N^{max.con.}$ reached 3000, the number of platform users and the profits of the logistics firms could acquire the largest values with almost no additional delivery fee. However, at $N^{max.con.} > 3000$, there is a difference of about 0.015 to about 0.022 between the $s^{logi.}$ for the maximization on the platform users' side and the $s^{logi.}$ for the maximization of the logistics firms' side. Therefore, the results indicate that platform owners had to trade-off between their platform user base size and the logistics firms' profits.

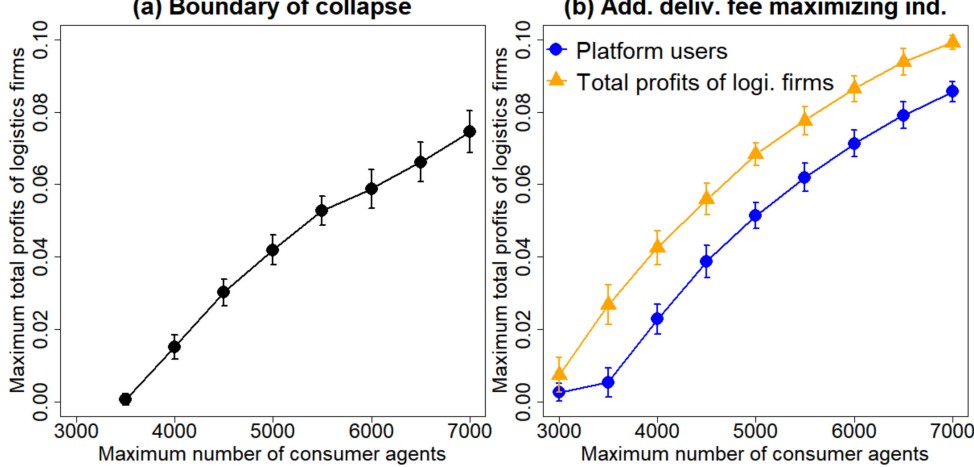

**Figure 7.** Trajectory of the boundary of collapse and additional delivery fees maximizing each indicator. We note that the value of logistics firms in figure (**b**) at $N^{max.con.} = 7000$ might be calculated to be smaller than the real value because of the maximum value of $s^{logi.}$ in these experimental settings.

### 3.2. Comparison before and after the Introduction of the Ecosystem Strategy

Figures 8 and 9 show the results of the comparison before and after the introduction of the ecosystem strategy. First, Figure 8 shows the results of the comparison by way of the trajectory of the boundary of collapse. The results demonstrated that the introduction of the strategy moved the trajectory in the direction of a lower additional delivery fee. The difference between the $s^{logi.}$ of the trajectories before and after was from about 0.030 to 0.034. Additionally, the results indicate that the introduction of the strategy provided postponement for collapse of the ecosystems: it was equivalent of platform users of about 2000 consumer agents (20 million consumers) at $s^{logi.} = 0.04$. Therefore, the results imply that the introduction of the strategy can decrease the risk of an ecosystem collapse. Thus, hypothesis (a) was supported.

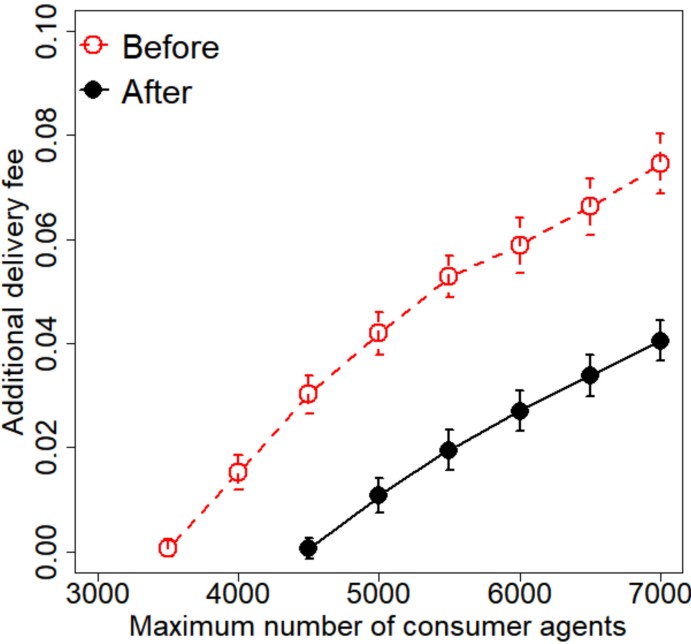

**Figure 8.** Comparison of the trajectory of the boundary of collapse. In the figure, "before" means before the introduction of the ecosystem strategy and "after" means after the introduction of the ecosystem strategy.

Second, Figure 9a,b shows the comparison of the trajectory of the $s^{logi.}$ maximizing platform users (indicator A) and the logistics firms' total profits (indicator B), respectively. Figure 9c shows the comparison of the maximum values of platform users (indicator A) at each $N^{max.con.}$ and Figure 9d shows the comparison of maximum values of the logistics firms' total profits (indicator B) at each $N^{max.con.}$. As the results indicate, both the number of platform users and logistics firms' total profits increased due to the introduction of the ecosystem strategy. The increase was observed after the collapse of the ecosystem was risked ($N^{max.con.} > 3000$). The results showed that the amount of increase grew larger as the $N^{max.con.}$ value increased. At $N^{max.con.} = 7000$, it increased the number of platform users to about 411 consumer agents (about 1.10 times) and increased total profits of the logistics firms by about 1016 (about 1.22 times). Thus, hypothesis (b) was supported. Additionally, as shown in Figure 9a,b, the trajectory maximizing the number of platform users and maximizing the profits of logistics firms almost corresponded after the introduction of the strategy, although there was a trade-off between the platform user base size and the logistics firms' profits before its introduction. Accordingly, the results implied that the introduction of the ecosystem strategy for physical intermediary firms allowed that the platform owners could maximize both of platform users and profits of logistics firms.

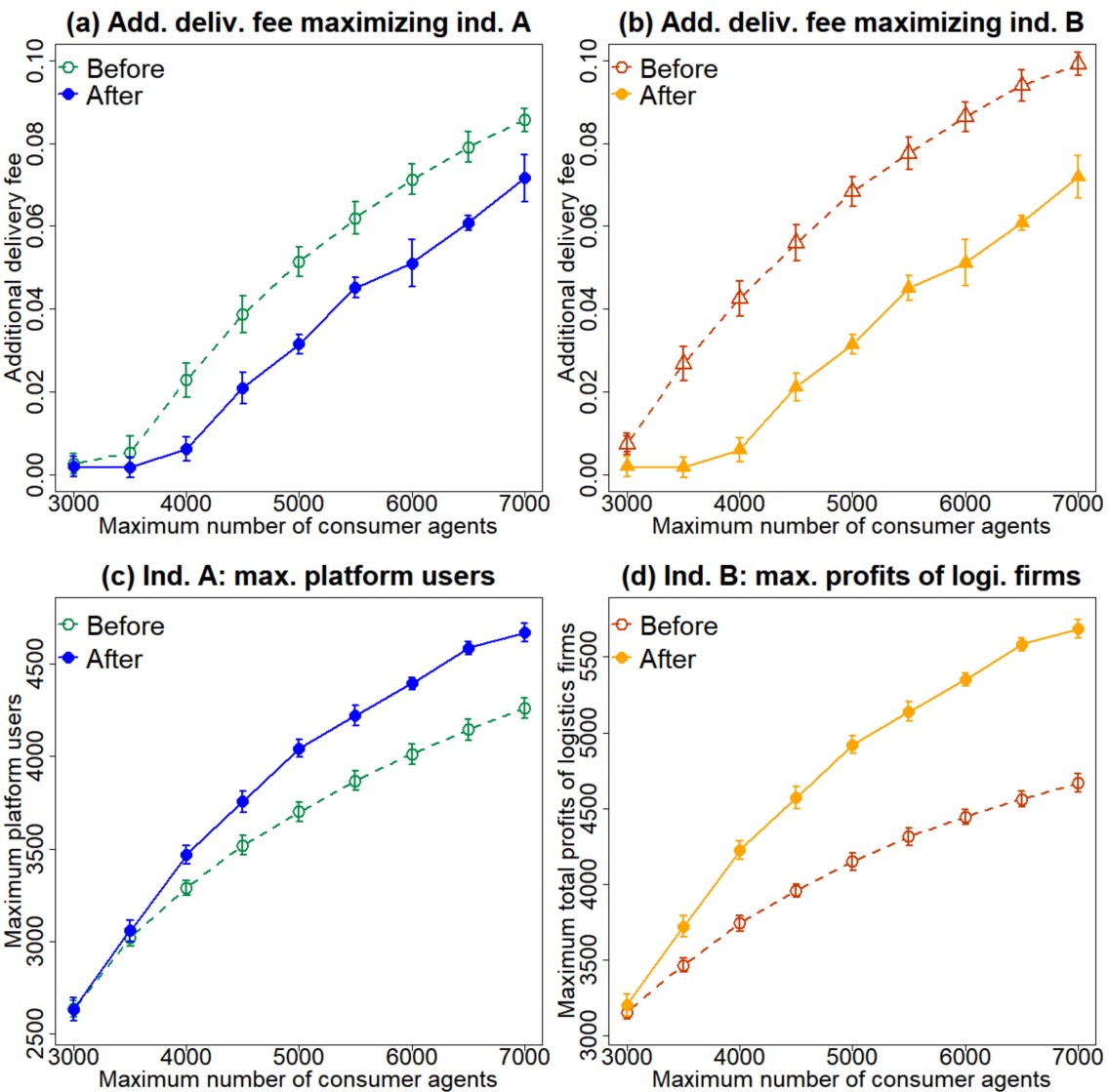

**Figure 9.** Comparison of the trajectory of additional delivery fees maximizing each indicator and the maximum value of each indicator. In the figure, "before" means before the introduction of the ecosystem strategy and "after" means after the introduction of the ecosystem strategy.

## 4. Discussion

This study focused on the collapse of ecosystems caused by physical intermediary firms. It aimed to test how the application of an ecosystem strategy in these firms can contribute to avoiding such collapse and facilitate the evolution and sustainability of platform-based markets. As an alternative approach to logistics management [16–18], this study developed an ecosystems strategy [19] to maintain the sustainability of a platform-based marketplace in terms of logistics firms. We constructed an agent-based simulation customized for Japanese platform-based marketplaces to test our hypothesis. The results indicate that the introduction of an ecosystem strategy postponed reaching the boundary of collapse and delayed the collapse of the ecosystem. Additionally, it increased the number of platform users by about 1.10 times and increased the total profits of logistics firms by about 1.22 times. Although these results cannot compare with the results of logistics management research, at the very least, we confirmed the effectiveness of the proposed ecosystem strategy. Additionally, we consider that it may be possible to adopt the ecosystem strategy in parallel with optimization based on logistics management. Therefore, this study illustrated that an ecosystem strategy for physical intermediary firms is effective

in the evolution of platform-based markets and its applications facilitate the sustainability of the markets by preventing the collapse of platform ecosystems.

In addition to these main results, we confirmed that our results indicated that the introduction of the ecosystem strategy eliminated the trade-off relationship between platform users and the profits of logistics firms and allowed for the maximization of both. In intermediary platforms such as the marketplace, since the platform owner needs to gain profit from either the supply or demand side [44,55,57], the structure of profits between these sides becomes the trade-off. Although a previous study indicated that intermediary platforms could increase the profits of both sides in certain types of markets [41], the trade-off relationship remained. Therefore, we believe our results have novelty, as they imply that certain ecosystem strategies could resolve this relationship. We inferred that this is caused by the following mechanisms: The logistics firm agent will decide to outsource work if the delivery demands exceed their delivery capacity. Here, there is a time lag between the occurrence of lack of capacity and commencement of outsourcing. This occurrence happens outside simulation settings too. Since delivery delays due to insufficient delivery capacity happen in the real world, we have to consider that logistics firms may find it difficult to prepare (or intentionally do not prepare) for a lack of capacity in advance. In such instances, when the proposed ecosystem strategy is introduced, a delivery delay due to an insufficient delivery capacity has a larger influence on consumers, since the delay occurs across all major logistics firms. We consider that this delay could automatically adjust the size of consumers' platform use. In contrast, before the introduction of the ecosystem strategy, a single major logistics firm can cause the delivery delay. This means that the influence on delivery delay is relatively small and it cannot control consumers' demands. Thus, our results imply that the introduction of ecosystem strategy, which facilitates standardization and cooperation among physical intermediary firms, benefits the platform ecosystem by generating a new self-adjustment mechanism among actors.

*4.1. Theoritical Implications*

This study provides three major implications. First, it demonstrates the importance of physical intermediary firms in platform ecosystems. Previous research on platform ecosystems has investigated platform owners, complementors as product providers, and consumers (e.g., [1,7,20–22,41]). However, as summarized in Table 1, the existence of physical intermediary firms is mandatory for accomplishing the value chain in platform ecosystems. We deem that relationship building between physical intermediary firms and platform owners becomes especially significant in the early and latter growth stages of the platform-based market. In the early stage, platform owners urge physical intermediary firms to achieve a sufficiently high service quality as a value proposition of the platform ecosystem. In the latter end of the growth stage, platform owners should prevent physical intermediary firms from failing to achieve the required service provisions in cases of excessive demand. Although a delivery crisis might only be observable in Japan at this time, the large growth of various platform-based markets raises several concerns, wherein similar situations might occur in various sectors and countries in the future. This study contributes to the research stream by providing the first steps in the investigation of physical intermediary firms in platform ecosystems.

Second, this study developed methods for adapting an agent-based simulation into predictions regarding the effectiveness of applying an ecosystem strategy on platform ecosystems. Boero and Squazzoni [67] suggest that empirical knowledge needs to be appropriately embedded into modeling practices through specific strategies and methods. Therefore, we attempted to reproduce a simulation environment that closely mimics reality by constructing a consumer decision-making model based on a questionnaire survey and mapping real address data for consumers and logistics bases. Although a theoretical agent-based simulation that is simpler or purer can provide clarification on the structure, such simulation methods risk potential changes due to changing parameters, which are untested by actual data. Therefore, we believe that our methods, based on real datasets, provide more reliable and valid simulation results. Furthermore, because the platform ecosystem can become more complex with

the interactions of various actors, we consider our approach effective. However, one disadvantage of our proposed approach is that it complicates the study procedure. Therefore, in the Appendix B, we attempt to simplify our proposed simulation model for future research.

Third, an implication of this study is about the expansion of sustainability research in platform ecosystems. Although the platform ecosystem includes an "ecosystem", its sustainability aspect is almost undeveloped. As a current example, Inoue and Tsujimoto [7,8] studied platform ecosystems focusing on new markets and analyzed its unsustainability caused by the profitability of complementors. Inoue, Takenaka, and Kurumatani [41] expanded these studies and simulated the sustainability of platform ecosystems in the service industry. Miron, Purcarea, and Negotia [76] investigated the entry of high-quality complementors as a significant factor in ecosystem sustainability. Wan, Cenamor, Parker, and Van Alstyne [77] surveyed the organizational ambidexterity of platform owners in terms of sustainability. This study expanded these research streams in terms of "unsustainability caused by physical intermediary firms" and "maintaining sustainability by introducing an ecosystem strategy".

Fourth, we consider the ecosystem strategy in this study can be regarded as a way facilitating service open innovation. In the simulation, this study set this strategy has two effects as follows: (a) the platform owner makes logistics firms declare their delivery capacity and provides a delivery allocation system to equally distribute the delivery orders based on that capacity, and (b) the platform owner provides a system that facilitates each logistics firm to co-use the delivery base with other logistics firms. These effects are realized by collaboration between the platform owner and logistics firms. Therefore, we deem such collaboration would be achieved as the processes of open innovation. For successful open innovation in the ecosystems, the platform's technology openness strategy, complex adaptive systems, and market responses stimulated by technology innovations are important [78,79]. The platform owner must establish the platform policies to foster a sustainable environment of the ecosystem for open innovation between the platform owner and complementors [80]. This study implies that these management for the successful open innovation is significant not only for complementors, but also for physical intermediary firms.

### 4.2. Practial Implications

This study provides two major managerial implications. First, it demonstrates the importance of dynamic pricing for delivery fees in marketplace platform ecosystems. As the simulation results in Figure 6 show, we confirmed that a difference of just 0.005–0.010 in delivery fees could exceed the withdrawal boundary of logistics firms from the platform, causing platform ecosystem collapse. Therefore, platform owners must focus on the profitability of logistics firms in the ecosystem and the delivery fee settings over time. Without this focus, it will not be possible to secure the sustainability of marketplace platform ecosystems. Additionally, as Figure 9 shows, since introducing the ecosystem strategy for physical intermediary firms could resolve the trade-off between platform users and profits of logistics firms, it allows the implementation of such dynamic pricing to retain platform users. Therefore, our results imply that the introduction of such ecosystem strategy can support the smooth implementation of dynamic pricing. We believe that platform owners autonomously change the delivery fee to maintain the sustainability of the ecosystem.

Second, we believe that our simulation methodology has value in terms of future practical applications. As our simulation is based mainly on actual data, the acquired results can be interpreted as an estimation of actual indicators, such as the number of platform users. As some of the indicators in this study are simplified or assumed, our simulation does not reach practical levels for referring the estimated values. However, future development of such simulation methods can obtain predictions that inform effective measures for the evolution and sustainability of platform ecosystems.

### 4.3. Limitations and Future Work

This study has several limitations. First, the analysis and simulation are specific to the Japanese market of marketplace platforms, limiting the applicability of this study's implications to the said

market. Therefore, future research should analyze and simulate other regions to understand the effect of logistics firms on platform ecosystems globally.

Second, for the sake of simplicity, this study restricted its focus to the delivery fee settings between consumers and logistics firms. However, we can suppose situations wherein the platform owner increases the platform fees of third-party sellers for additional delivery fees in order to assure consumers' profits, whether or not the settings actually exist or not. Additionally, some platform owners may choose to bear additional delivery fees themselves. Therefore, future research could study the broader fee setting patterns to evolve the platform ecosystems.

Third, this study focuses on delivery fees, which is the most basic measure related to the participation of logistics firms. However, there are various other measures that platform owners can implement regarding logistics companies, such as the establishment of delivery boxes, delivery by platform owners themselves, and the encouragement of delivery acceptance by consumers at logistics firms' delivery centers or free depots. Additionally, such novel approaches could allow for the implementation of a new ecosystem strategy. Thus, future studies could verify how these measures can support logistics firms and improve the evolution and sustainability of platform ecosystems.

## 5. Conclusions

This study tested how the application of an ecosystem strategy on physical intermediary firms can contribute to circumventing the collapse of platform ecosystems and facilitating the evolution and sustainability of the platform-based market. We elaborated the ecosystem strategy in this study as "the strategy that the platform owner uses to cooperate with logistics firms to achieve standardization of logistics services and improves platform system cooperation among them". We then constructed an agent-based simulation system using a dataset of Japanese platform-based marketplace and tested the effectiveness of this ecosystem strategy. The results confirmed that the application of an ecosystem strategy for physical intermediary firms is effective both in avoiding the collapse of platform ecosystems and in improving the profits of actors within the ecosystem. In addition, our results indicated that the introduction of this strategy eliminated the trade-off relationship between platform users and the profits of logistics firms and allowed for the maximization of both. Thus, this study expanded the research on platform ecosystems and its sustainability by verifying the effectiveness of the strategy for ecosystem evolution and sustainability. This study has three major limitations: its analysis is limited to the Japanese marketplace, restricted pricing scheme between logistics firms and consumers, and it tests specific content of an ecosystem strategy. Therefore, future research could expand this study in three directions: focusing on the marketplace platforms of several countries, pricing scheme including third-party sellers and platform owners, and testing the effectiveness of other contents of ecosystem strategy.

**Author Contributions:** Conceptualization, Y.I., M.H. and T.T.; methodology, Y.I., M.H. and T.T.; software, Y.I.; validation, Y.I.; formal analysis, Y.I.; investigation, Y.I. and M.H.; resources, Y.I. and M.H.; data curation, Y.I.; writing—original draft preparation, Y.I. and M.H.; writing—review and editing, Y.I., M.H. and T.T; visualization, Y.I.; supervision, Y.I.; project administration, Y.I.; funding acquisition, Y.I.

**Funding:** This work was funded by JSPS KAKENHI Grant Numbers 18K12874.

**Conflicts of Interest:** The authors declare no conflict of interest.

## Appendix A. Detailed Specifications of Agent Designs

*Appendix A.1. Consumer Agents*

In this supplementary section, we explain the specifications of consumer agents in more detail.

Appendix A.1.1. Settings of Consumer Agents

Consumer agents—included as $N^{consumer}$—make decisions on whether to use the platform or not, respectively. Each consumer had the following parameters:

$\beta_i^1$: Influence of the platform usage fee on platform use;

$\beta_i^2$: Influence of the product variety on the platform on platform use;

$\beta_i^3$: Influence of the delivery delay probability on the platform;

$\beta_i^4$: Influence of the minimum delivery period on platform use;

$f_i^{Frequency}$: Probability of considering platform use per simulation step;

$f_i^{TimeSpecif.}$: Probability of appointing a day of the week and time for product delivery;

$f_i^{Acce.Time}$: The day of the week and the time when the agent can receive products at home;

$f_i^{place}$: Latitude and longitude of the home location of the consumer agent.

We set the values of each of these items according to the survey results. Each value was randomly determined for each agent, based on the distribution of values from analytical results or datasets. The days of the week ranged from Monday to Sunday and the time periods were categorized as morning (09:00 to 12:00), afternoon (12:00 to 15:00), evening (15:00 to 18:00), and night (18:00 to 21:00). Defining $q_t^{deli.}$ as the minimum number of days from the time of order at step $t$ on the platform to delivery at a consumer's home, the consumer agent can appoint the delivery date and time after $t + q_t^{deli.}$. When the consumer agent does not appoint the delivery date and time, the platform agent sets the date and time at step $t + q_t^{deli.}$ and a random time (random hours). The value of $q_t^{deli.}$ is set based on the circumstances of the delivery on the platform (details are described in the subsection of the logistics firm agents). This simulation set the minimum value of $q_t^{deli.} = 2$.

We set the values of $f_i^{place}$ based on the geographical population dataset. We collected the latitude and longitude values of Japanese streets and the $f_i^{place}$ values were randomly set with the probability based on the population of each street. Figure A1 shows the latitude and longitude values of our dataset. These data were obtained from e-Stat, the website of Japanese Government Statistics [74]. In our simulation, we excluded parts of streets to which delivery was difficult; streets that were 20 km or more away from the closest delivery base of any logistics firms were excluded (about 1% of all streets).

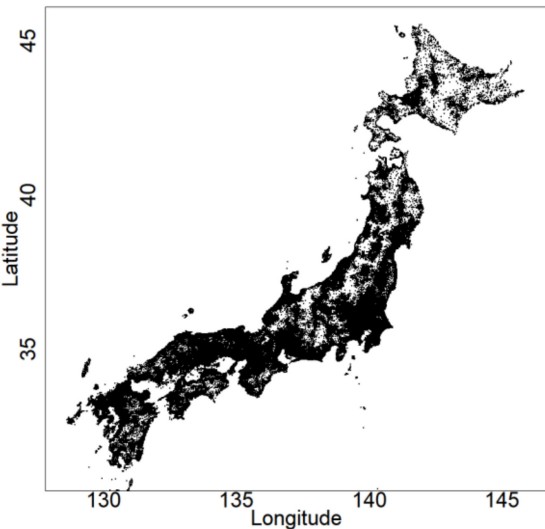

**Figure A1.** Latitude and longitude values of Japanese streets. This figure excludes some islands for visual clarity.

Appendix A.1.2. Consumer Questionnaire Survey

(A)  Sampling

The respondents of this survey were provided by Macromill Co., Ltd., which is one of the largest internet research companies in Japan. The pool of Macromill's survey respondents amounts to 1.2 million (at 11 April 2019) and covers a wide range of ages. We asked for 2000 respondents. Our survey started on 12 November 2018, and ended on 13 November 2018, after the expected number of responses was acquired. Finally, we obtained 2060 valid responses. The ages of the respondents ranged from 20 years old to 85 years old.

As an initial screening of respondents, we tried to extract consumers that often used products purchased from delivery platforms for personal use (not for business tasks). In order to restrict the respondents, we set the following two screening questions: Q1. How often do you use marketplace platforms? and Q2. What kind of products do you purchase on marketplace platforms? In response to these questions, we chose respondents who answered "at least once a month" for Q1 and "products for personal and daily goods" for Q2. By this screening, we lost potential respondents who use rarely marketplace platforms. However, this influence is small, since consumers who actively use such platforms would influence the dynamics of platform ecosystems more than those potential respondents would. We obtained 2060 valid responses after this screening process.

(B)  Questionnaire Design

The questionnaire survey had two parts. The first part collected information on the current usage of marketplace platforms. We utilized the following information in our simulation from this part: (a) the frequency of marketplace platform usage per month, (b) the probability of appointing a delivery date and time, (c) the day of the week and time of the day when the product can be received at home, and (d) the evaluation of the degree of product variety on marketplace platforms compared to that in retail shops.

The second part collected information to build the decision models for marketplace platform usage. In this part, we conducted a conjoint analysis. We asked questions based on a hypothetical situation, namely "When you buy some products that are worth 5000 Japanese yen (about 45–50 US dollars), do you use marketplace platforms with the following conditions or a retail store?" We then presented them with a series of conditions consisting of the following combinations of elements, including four factors and five levels.

- Payment amount: (1000 yen discount, 500 yen discount, price the same as the retail store, 500 yen higher, 1000 yen higher).
- Product variety: (1/4 of the store's, 1/2 of the store's, same level as the store, double the store's, four times the store's).
- Probability of delay in delivery: (0%, 25%, 50%, 75%, 100%).
- Minimum delivery period: (0 days, 3 days, 6 days, 9 days, 12 days).

Because we used an orthogonal array, the survey included 25 question patterns presented in random order.

(C)  Analysis, and Building Consumer Decision Models

Setting the probability that the consumer $i$ selects the platform as $p_i$, the platform usage fee for the consumer as $q^{pay}$, the product variety on the platform as $q^{vari.}$, the delay probability as $q^{delay}$, and the minimum delivery period as $q^{deli.}$, we can express the decision model of consumers for platform usage as:

$$\ln \frac{p_i}{1 - p_i} = \beta_i^1 q^{pay} + \beta_i^2 \ln q^{vari.} + \beta_i^3 q^{delay} + \beta_i^4 q^{deli.} + C_i, \tag{A1}$$

where $C_i$ is the basic degree of use of the platform when all other elements are zero. Based on the survey responses, its logarithm is calculated as the product variety $q^{vari.}$ is captured as the ratio product variety

on the platform in comparison with a retail store. The platform usage fee $q^{pay}$ converts from the valued obtained from the answer items of the questionnaire (1000 yen discount, 500 yen discount, price the same as the retail store, 500 yen higher, 1000 yen higher) as ($-0.2, -0.1, 0, 0.1, 0.2$) for a generalization that depends on the price of the product (dividing the values of the response from each question item by 5000 Japanese yen). Negative values of $q^{pay}$ imply a discount for consumers as an incentive.

Procedure A. In this procedure, we defined the response patterns of each respondent. Since the number of data samples of each respondent was 25, the application of the logistic model of Equation (A1) was difficult. Instead, we calculated a test for noncorrelation between the answer of platform usage (the value is one when the answer is that they make use of marketplace platforms and the value is zero when the answer is they make use of retail stores) and the value of each factor, namely the $q^{pay}$, $\ln q^{vari.}$, $q^{delay}$, and $q^{deli.}$, of each consumer. If the factor has a significance value of $p < 0.05$ for platform use, the factor influences the consumer's platform use. Here, our classification was applied only to coefficients that made semantic sense. Accordingly, significance was considered if the coefficient with $q^{pay} < 0$, coefficient with $\ln q^{vari.} > 0$, coefficient with $q^{delay} < 0$, and coefficient with $q^{deli.} < 0$ are satisfied in each instance. In this calculation, we used Spearman's nonparametric rank correlation.

Procedure B. In this procedure, we classified the consumers based on the results from Procedure A simply by patterns of influence factors. For example, if the influential pattern of a consumer is (influence of $q^{pay}$, influence of $\ln q^{vari.}$, influence of $q^{delay}$, influence of $q^{deli.}$ = yes, yes, no, no), the consumer is included in the group where all consumers have a (yes, yes, no, no) pattern.

Procedure C. In this procedure, we performed a logistic regression analysis by applying Equation (A1) to each consumer group to estimate the coefficient values and standard errors of each factor. This study acquired the estimation results for each classification and applied them to the decision model of consumer agents in our simulation at a rate that corresponds to the sample rate of each group.

(D) Consumer Questionnaire Survey Results

The gender breakdown of the respondents of this survey was 48.9% male and 51.1% female. In terms of age, 11.3% were in their 20s, 22.6% in their 30s, 25.5% in their 40s, 23.0% in their 50s, and 17.5% in their 60s or older. To overcome a potential bias in these rates, we calculated the correction weight for each gender and age based on the actual population distribution data of internet users at the application for the simulation. This correction method was applied to all of the results related to the questionnaire survey.

Figure A2 shows the distribution of the frequency of marketplace platform use. The results showed that some rate of consumers frequently use platforms, although a major rate of consumers uses them 1~3 times per month. We translated these survey items to numerical values as usage rates per day and set these as probability of considering platform use $f_i^{Frequency}$ based on their occurrence probability. The mean value of $f_i^{Frequency}$ was about 0.13.

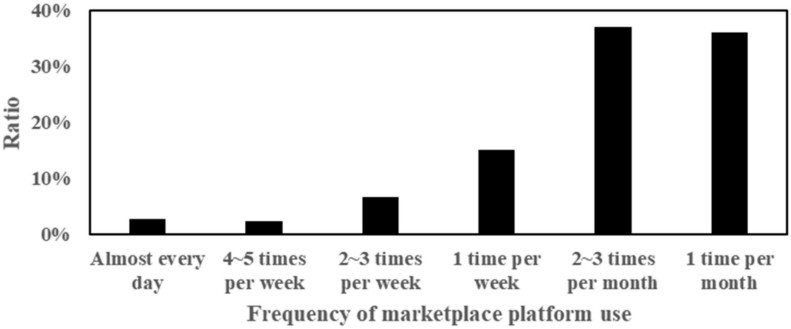

**Figure A2.** Distribution of the frequency of marketplace platform use.

Figure A3 shows the distribution of the probability of a consumers' appointing a delivery date and time. The results show that about 35% of consumers do not appoint a date and time, while about 20% almost certainly do. As the expected value for the appointment is 34.5%, about 65.5% of deliveries risk redelivery. We applied these distributions of probability to $f_i^{TimeSpecif.}$ of the consumer agents.

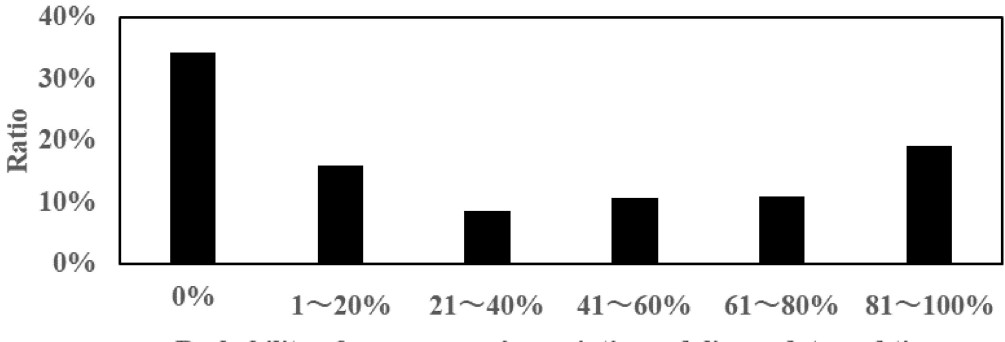

**Figure A3.** Distribution of probability for consumers' appointment of a delivery date and time.

Figure A4 shows the classified patterns of delivery acceptance dates and times of the consumers. We classified these responses by the hierarchical clustering of Ward's method, which resulted in 19 classes, all of which have more than one percent of respondents. We rounded the values in this table off to the nearest integers for simplicity. The largest class was made up of consumers who could receive deliveries at home at all times, including housewives and retirees. The second-most common class was that of consumers who could receive deliveries only during the night, mainly including working people. The third class was that of consumers who could only receive deliveries in the morning, mainly including part-time workers and the self-employed. We assigned these 19 classes to each consumer agent according to probabilities of occurrence and set values of $f_i^{Acce.Time}$.

| Class | 1 | 2 | 3 | 4 | 5 | 6 | 7 | 8 | 9 | 10 | 11 | 12 | 13 | 14 | 15 | 16 | 17 | 18 | 19 |
|---|---|---|---|---|---|---|---|---|---|---|---|---|---|---|---|---|---|---|---|
| Rate (%) | 18 | 14 | 8 | 8 | 6 | 6 | 6 | 4 | 4 | 4 | 4 | 3 | 3 | 3 | 3 | 2 | 2 | 1 | 1 |
| Mon. 09:00 - 12:00 | 1 | 0 | 1 | 0 | 0 | 0 | 0 | 1 | 0 | 1 | 0 | 0 | 1 | 0 | 1 | 0 | 1 | 0 | 1 |
| 12:00 - 15:00 | 1 | 0 | 0 | 0 | 0 | 0 | 0 | 1 | 0 | 0 | 0 | 0 | 1 | 1 | 1 | 1 | 1 | 1 | 0 |
| 15:00 - 18:00 | 1 | 0 | 0 | 0 | 0 | 0 | 0 | 1 | 1 | 0 | 1 | 0 | 0 | 1 | 1 | 0 | 0 | 1 | 1 |
| 18:00 - 21:00 | 1 | 1 | 0 | 1 | 1 | 1 | 1 | 0 | 1 | 1 | 0 | 0 | 1 | 1 | 1 | 0 | 0 | 0 | 0 |
| Tue. 09:00 - 12:00 | 1 | 0 | 1 | 0 | 0 | 0 | 0 | 1 | 0 | 1 | 0 | 0 | 0 | 0 | 1 | 0 | 1 | 0 | 1 |
| 12:00 - 15:00 | 1 | 0 | 0 | 0 | 0 | 0 | 0 | 1 | 0 | 0 | 0 | 0 | 1 | 1 | 1 | 1 | 1 | 1 | 0 |
| 15:00 - 18:00 | 1 | 0 | 0 | 0 | 0 | 0 | 0 | 1 | 1 | 0 | 1 | 0 | 0 | 1 | 1 | 0 | 0 | 1 | 1 |
| 18:00 - 21:00 | 1 | 1 | 0 | 1 | 1 | 1 | 1 | 0 | 1 | 1 | 0 | 0 | 1 | 1 | 1 | 0 | 0 | 0 | 0 |
| Wed. 09:00 - 12:00 | 1 | 0 | 1 | 0 | 0 | 0 | 0 | 1 | 0 | 1 | 0 | 0 | 0 | 0 | 1 | 0 | 1 | 0 | 1 |
| 12:00 - 15:00 | 1 | 0 | 0 | 0 | 0 | 0 | 0 | 1 | 0 | 0 | 0 | 0 | 1 | 1 | 1 | 1 | 1 | 1 | 0 |
| 15:00 - 18:00 | 1 | 0 | 0 | 0 | 0 | 0 | 0 | 1 | 1 | 0 | 1 | 0 | 0 | 1 | 1 | 0 | 0 | 1 | 1 |
| 18:00 - 21:00 | 1 | 1 | 0 | 1 | 1 | 1 | 1 | 0 | 1 | 1 | 0 | 0 | 1 | 1 | 1 | 0 | 0 | 0 | 1 |
| Thu. 09:00 - 12:00 | 1 | 0 | 1 | 0 | 0 | 0 | 0 | 1 | 0 | 1 | 0 | 0 | 0 | 0 | 1 | 0 | 1 | 0 | 1 |
| 12:00 - 15:00 | 1 | 0 | 0 | 0 | 0 | 0 | 0 | 1 | 0 | 0 | 0 | 0 | 1 | 1 | 1 | 1 | 1 | 1 | 0 |
| 15:00 - 18:00 | 1 | 0 | 0 | 0 | 0 | 0 | 0 | 1 | 1 | 0 | 1 | 0 | 0 | 1 | 1 | 0 | 0 | 1 | 1 |
| 18:00 - 21:00 | 1 | 1 | 0 | 1 | 1 | 1 | 1 | 0 | 1 | 1 | 0 | 0 | 1 | 1 | 1 | 0 | 0 | 0 | 0 |
| Fri. 09:00 - 12:00 | 1 | 0 | 1 | 0 | 0 | 0 | 0 | 1 | 0 | 1 | 0 | 0 | 0 | 0 | 1 | 0 | 1 | 0 | 1 |
| 12:00 - 15:00 | 1 | 0 | 0 | 0 | 0 | 0 | 0 | 1 | 0 | 0 | 0 | 0 | 0 | 1 | 1 | 1 | 1 | 1 | 0 |
| 15:00 - 18:00 | 1 | 0 | 0 | 0 | 0 | 0 | 0 | 1 | 1 | 0 | 1 | 0 | 0 | 1 | 1 | 0 | 0 | 1 | 1 |
| 18:00 - 21:00 | 1 | 1 | 0 | 1 | 1 | 1 | 1 | 0 | 1 | 1 | 0 | 0 | 1 | 1 | 1 | 0 | 0 | 0 | 1 |
| Sat. 09:00 - 12:00 | 1 | 0 | 1 | 1 | 0 | 1 | 1 | 1 | 0 | 1 | 0 | 0 | 0 | 0 | 1 | 0 | 1 | 0 | 1 |
| 12:00 - 15:00 | 1 | 0 | 0 | 1 | 0 | 0 | 0 | 1 | 0 | 0 | 0 | 0 | 1 | 0 | 1 | 1 | 1 | 1 | 0 |
| 15:00 - 18:00 | 1 | 0 | 0 | 1 | 1 | 0 | 0 | 1 | 1 | 0 | 1 | 0 | 1 | 0 | 1 | 0 | 0 | 0 | 1 |
| 18:00 - 21:00 | 1 | 1 | 0 | 1 | 1 | 0 | 1 | 0 | 1 | 1 | 0 | 1 | 1 | 1 | 1 | 0 | 0 | 0 | 1 |
| Sun. 09:00 - 12:00 | 1 | 0 | 1 | 1 | 0 | 1 | 1 | 1 | 0 | 1 | 0 | 0 | 0 | 0 | 1 | 0 | 1 | 0 | 1 |
| 12:00 - 15:00 | 1 | 0 | 0 | 1 | 1 | 0 | 0 | 1 | 0 | 0 | 1 | 0 | 1 | 0 | 1 | 1 | 1 | 1 | 0 |
| 15:00 - 18:00 | 1 | 0 | 0 | 1 | 1 | 0 | 0 | 1 | 1 | 0 | 1 | 0 | 1 | 0 | 1 | 0 | 0 | 1 | 0 |
| 18:00 - 21:00 | 1 | 1 | 0 | 1 | 1 | 0 | 1 | 0 | 1 | 1 | 0 | 1 | 1 | 1 | 1 | 0 | 0 | 0 | 1 |

**Figure A4.** Classified patterns of delivery acceptance dates and times. In these cells, 1 means the consumer is at home and 0 means the consumer is absent.

Table A1 shows the results of the consumer decision model related to platform use. We adopted six models, all of which included at least one percent of respondents. These decision models were assigned to consumer agents according to their rate of consumers. Model 1 included the largest number of consumers, accounting for 73% of the total. Applying Model 1 revealed that a consumer's decision to use the platform is largely influenced by the platform usage fee and discount rates. Model 2 included 11% of the consumers. Applying Model 2 revealed that a consumer's decision to use the platform is largely influenced by the length of delivery time, as well as the platform usage fee and discount rates. Model 3 included 8% of the consumers. Applying Model 3 revealed that a consumer's decision to use the platform is largely influenced by the length of delivery time. Although Models 4, 5, and 6 also represented distinct features, these models only represented about 7% of the consumers. Therefore, the results indicate that about 85% of Japanese marketplace platform users are motivated by lower prices.

**Table A1.** Consumer decision-making models regarding platform use

| | Model 1 | | | Model 2 | | | Model 3 | | |
|---|---|---|---|---|---|---|---|---|---|
| Variables | Value | S.E. | *p* Value | Value | S.E. | *p* Value | Value | S.E. | *p* Value |
| Influence of platform usage fee | −17.85 | 0.19 | ** | −15.87 | 0.51 | ** | −5.47 | 0.38 | ** |
| Influence of product variety on the platform | 0.15 | 0.02 | ** | 0.14 | 0.05 | ** | 0.25 | 0.05 | ** |
| Influence of delay probability of delivery | −0.78 | 0.05 | ** | −1.14 | 0.14 | ** | −0.09 | 0.15 | |
| Influence of minimum delivery period | −0.09 | 0.00 | ** | −0.48 | 0.02 | ** | −0.44 | 0.02 | ** |
| Intercept | 0.38 | 0.04 | ** | 1.66 | 0.11 | ** | 0.86 | 0.10 | ** |
| Rate of consumer | 0.73 | | | 0.11 | | | 0.08 | | |
| | Model 4 | | | Model 5 | | | Model 6 | | |
| Variables | Value | S.E. | *p* Value | Value | S.E. | *p* Value | Value | S.E. | *p* Value |
| Influence of platform usage fee | −5.94 | 0.66 | ** | −16.34 | 1.24 | ** | −3.10 | 0.64 | ** |
| Influence of product variety of the platform | 0.46 | 0.10 | ** | 0.37 | 0.13 | ** | 1.23 | 0.11 | ** |
| Influence of delay probability of delivery | −4.63 | 0.32 | ** | −5.97 | 0.46 | ** | −0.39 | 0.26 | |
| Influence of minimum delivery date | −0.11 | 0.02 | ** | −0.23 | 0.03 | ** | −0.04 | 0.02 | * |
| Intercept | 0.71 | 0.17 | ** | 1.95 | 0.30 | ** | −0.39 | 0.20 | * |
| Rate of consumer | 0.03 | | | 0.02 | | | 0.02 | | |

Note. S.E. refers to standard errors. *: $p < 0.05$, **: $p < 0.01$.

Figure A5 shows the expected platform use rate depending on the values of influential factors, given that other values of influential factors are zero. Here, we assumed a total of 1000 consumers and applied the models shown in Table A1 to the consumers, considering the rate of applied consumers. This figure shows that consumers are drastically influenced by changes in platform usage fee and discount rates. Conversely, product variety did not significantly motivate platform use. This result indicates that the indirect network effect for consumers from third-party sellers is relatively small in the Japanese platform-based markets. The results also show that service quality, including delivery delay and delivery time, influence the platform use of consumers. As delivery time can increase to infinity, an excessively low-quality delivery service could negate the benefits of high platform discount rates and high platform product variety.

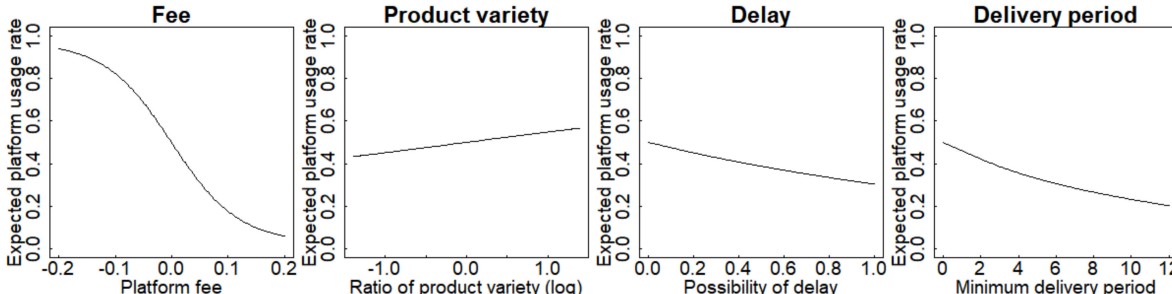

**Figure A5.** Expected platform use rate depending on values of influential factors given that other values of influential factors are zero. The x-axis of product variety is shown as a logarithm, that is, the range of original values is from 0.25 to 4.

*Appendix A.2. Logistics Firm Agents*

In this supplementary section, we explain the specifications of logistics firm agents in greater detail. To calculate the delivery delay and delivery time at each simulation step, we must calculate the number of consumer agents that each logistics agent can successfully deliver to. To do so, we must calculate the delivery time needed for each consumer agent. In this study, this includes the delivery time it takes to reach each consumer and the moving times between the delivery base and the district of the consumer agent.

First, we need to explain the calculation of delivery times among consumers' homes. Based on the pre-interviews with Yamato employees, this study assumed that the average delivery time for each consumer is 5.5 min from the delivery vehicle to the consumers' home entrance and back again. For simplicity's sake, we applied this value to all delivery cases in this simulation. Therefore, to handle 10,000 deliveries (i.e., one delivery for one consumer agent), logistics firm agents must spend approximately 917 h (5.5 ÷ 60 × 10,000) on the job at least.

Second, we will explain the calculation of moving times between the delivery bases and the districts of the consumer agents. This study classified the times of day for three months as morning (09:00–12:00), afternoon (12:00–15:00), evening (15:00–18:00), and night (18:00–21:00). As described above, when logistics firms need to spend an average of 5.5 min on each consumer, each delivery vehicle can handle about 32.72 consumers every 180 min. Since our simulation set one consumer agent to mean 10,000 consumers, we assumed that 306 vehicles (÷ (180 ÷ 5.5)) are needed for each consumer agent. For simplicity's sake, we calculate the total moving time for one consumer agent as 306 × the shortest moving time (i.e., the moving time between nearest the delivery base of the logistics agent and the district of the consumer agent) × 2 (meaning from the vehicle to the customer's house and back again).

The delivery time to the street of the consumer agent was defined as $d_{i,x}/h$, where $d_{i,x}$ is the distance from the nearest delivery base to the street address and $h$ is the mean speed of the delivery vehicle. We calculated $d_{i,x}$ by (a) calculating the length between the coordinates of the base and the consumer's street and (b) converting the length as a unit of km by Hubeny's formula. This study assumed $h$ as 20 km/h. Figure A6 shows the distance $d_{i,x}$ for each street by major logistics companies as a gradation. As seen in this figure, we collected information on each delivery base from the websites of every logistics firms and translated them into coordinate data of longitude and latitude. As shown in this figure, $d_{i,x}$ decreases as the number of delivery bases increases. In the simulation, we set the value of $d_{i,x}$ for Others uniformly as 20 km, the same as that of $h$.

Assuming $n_{t,y,x}$ is a consumer agent group assigned to logistics company agent $x$ in step $t$ at time period $s$, the total minimum required time $m_{t,s,x}$ of delivery can be calculated as:

$$m_{t,s,x} = \sum_{i}^{n_{t,s,x}} \left( 917 + 712 \frac{d_{i,x}}{h} \right). \tag{A2}$$

Although we could conduct more precise delivery simulations, this study simplified the calculation of delivery time in order to reduce computational time. As our simulation designated 3 h time periods for delivery, each logistics firm agent $x$ could deliver $m_{t,s,x} \div (3 \times g_x)$ of consumer agents based on their maximum number of delivery vehicles $g_x$. Here, we rounded down the value of $m_{t,s,x}/3g_x$ to the nearest integer.

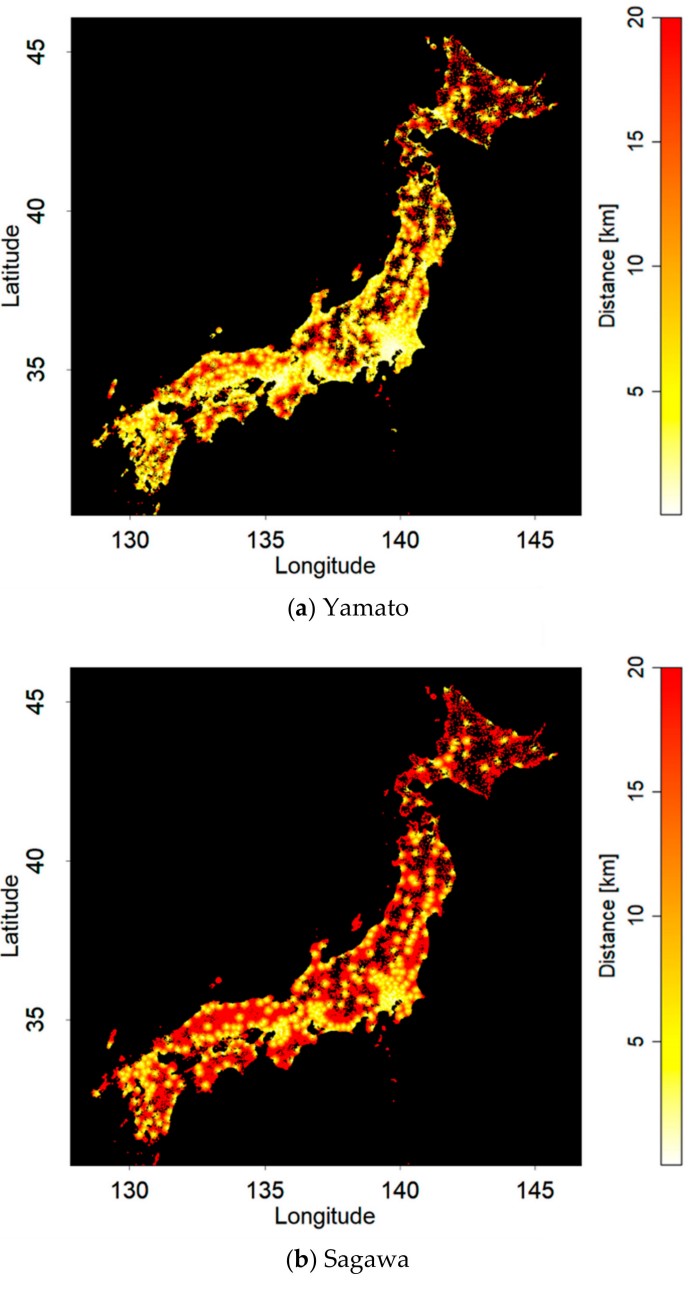

(**a**) Yamato

(**b**) Sagawa

**Figure A6.** *Cont.*

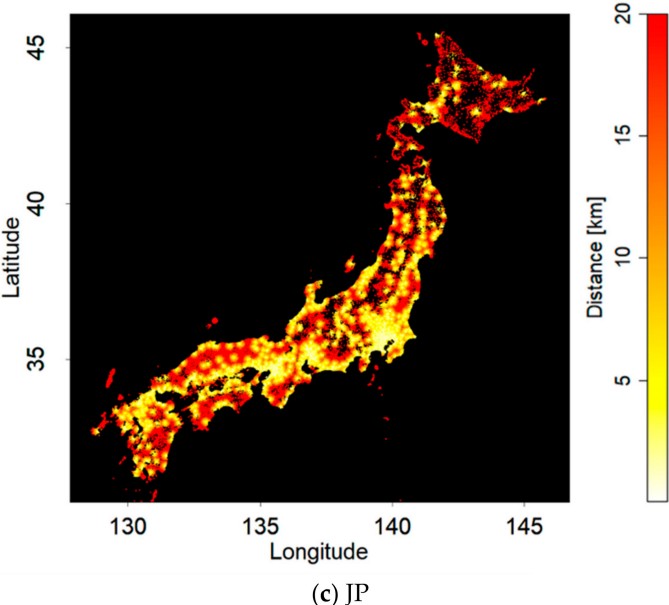

(**c**) JP

**Figure A6.** Distance between each consumer street address and the nearest delivery base of the logistics firm. The values of distance are shown as a gradation. The dot locations correspond to the coordinates of each Japanese street.

If the logistics firm agent could not fulfill the assigned delivery (caused by demands that exceed the upper limit of the agent or caused by the absence of the consumer at home), the rest of the delivery would be reassigned to the next simulation step. If the consumer agent could not receive the ordered product, the consumer agent would appoint an appropriate date and time for redelivery (such a re-delivery system is standard in Japan). The delay probability $q^{delay}$ was calculated for each consumer agent. That is, $q_i^{delay}$ is appropriate in our simulation. If the consumer agent could not receive the products at the appointed date and time, then $q_i^{delay}$ increased by a value of 0.2. If the consumer agent could receive products at the appointed step and time, then $q_i^{delay}$ decreased by a value of 0.2, where $q_i^{delay}$ ranges from 0 to 1.

The simulation system calculated the minimum delivery date $q^{deli.}$ by the delivery circumstances of logistics firm agents. The value of $q^{deli.}$ was calculated as the sum of the base delivery steps and the average difference between the initially-scheduled delivery step $t'$ and the actual delivery step $t''$ over the last 30 days. As the current mean delivery period in Japan is about two days, we set the base delivery steps as two.

In this simulation, logistics firm agents can increase their delivery capacity by obtaining an additional delivery fee, $s^{logi.}$.

Although we can suppose various ways to express increase of delivery capacity, we directly increase the simplest indicator of delivery capacity in our simulation: i.e., the number of delivery vehicles $g_x$. As the current Amazon.com delivery fee in Japan is 400 yen when the product price is less than 2000 yen, this simulation set the base delivery fee at 20% of the product price. As this simulation considers product price as constant, logistics firm agents can increase the number of their delivery vehicles $g_x$ by $(s^{logi.} + 0.2)/0.2$ times.

Finally, we will describe the revenue model and the withdrawal decision by logistics firm agents. The simulation assumed that each logistics firm agent considers using other firms' vehicles when the demand of delivery exceeds the upper limit of the number of their own delivery vehicles. This consideration was performed at every 30th step and the number of additional vehicles was calculated as $\hat{g}_x = \overline{m}_{t,s,x}/3 - g_x$, where $\overline{m}_{t,s,x}$ is the mean of delivery orders of the last 30 steps. This study simply assumed that logistics firm agents acquire 10 for each of their own deliveries and gain $-20$ (lose 20)

for each outsourced delivery. If the cumulative profit of logistics firm agent $x$ in the last 90 steps (meaning one quarter) is negative, the agent $x$ withdraws from the platform. Here, logistics firm agents represented by Others do not take any action in terms of outsourcing or withdrawal and they remain on the platform until the end of the simulation.

*Appendix A.3. Third-Party Seller Agents*

In this supplementary section, we explain the specification of third-party seller agents in more detail. Our simulation included third-party seller agents to calculate the product variety $q^{vari.}$ on the platform. Defining $j_t$ as the size of product provision by third-party seller agents, the indirect network effect from consumers can simply be expressed as

$$j_t = b\left(1 - fee^{TPS}\right)\frac{n_t}{N^{consumer}} \tag{A3}$$

where $fee^{TPS}$ is the platform usage fee for third-party sellers for each transaction, $n_t$ is the number of consumers using the platform, and $b$ is the constant of scale of product provision. As mentioned in the subsection on platform agents, this simulation set $fee^{TPS}$ as 0.1.

Then, setting $r$ as the size of product provision by third-party seller agents in retail shops, the degree of product variety on the platform can be expressed as

$$q^{vari.} = \frac{j_t}{r}. \tag{A4}$$

As stated in the subsection on consumer agents, the current number of Japanese Amazon.com users is approximately 40 million. As described in Section 2.5.1, the number of current internet shopping users in Japan is at least 70 million. Assuming that almost all platform users in Japan use Amazon.com regardless of their single-homing or multihoming preferences and that all internet users can be potential users of marketplace platforms, the current platform use rate can be calculated as 4/7. Additionally, we confirmed through our questionnaire survey for platform users of consumers that the average value of consumers' recognition of the degree of product variety on the platform was 1.5 times higher in comparison with that of retail shops. Therefore, we calculated $r/b$ as $(4/7)/1.5$, namely 0.381. Finally, integrating the value of $r/b$ and Equation (A3) into Equation (A4), the degree of product variety of the platform can be expressed as

$$q^{vari.} = 2.36\, n_t / N^{max.cons.}. \tag{A5}$$

**Appendix B. Simplification of the Simulation System**

This study tried to precisely reproduce Japanese marketplace platform ecosystems as an agent-based simulation. Although we believe this trial provides more reliable simulation results, it also complicates the procedure of this study. Therefore, in this Appendix, we consider factors that we can simplify to make it easier for future studies to utilize our proposed approach.

First, we summarize all parameters of consumer agents as mean values (and standard deviation values). For example, consumers' decision-making models, as in Table A1, can be summarized as an averaged decision model, shown in Figure A5. However, since this simplification ignores the distribution of each parameter, the obtained results will be different from the full simulation model, if the distributions of the parameters depart significantly from normal distribution.

Second, we may simplify the process of individual delivery for each consumer agent by calculating the mean value of minimum delivery time based on each delivery base (as shown in Figure A6) and the locations of each consumer agent. However, since this ignores the detailed process of delivery, the calculated values of delivery delay time and delay possibility may differ from the full model.

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
