# Peer review of "Effectiveness of Ecosystem Strategies for the Sustainability of Marketplace Platform Ecosystems"

_sustainability, doi:10.3390/su11205866_

Round 1
Reviewer 1 Report
Thank you for being given the opportunity to review this manuscript (“Effectiveness of ecosystem strategies for sustainability of product delivery platform ecosystems”), which I found to be a very interesting read. In their research, the authors report results from an agent-based simulation study, modeling a platform-based business with multiple companies for (physical) product delivery (i.e., parcel services) and consider how increasing load will eventually lead those to leave the market, and hence the overall market to collapse. Against the backdrop of the observation of this exact scenario from Japan, they evaluate how a “ecosystem strategy” implemented by the platform operator, can contribute to delay market collapse.
Overall, the paper reads very well and it is to note that extensive efforts have been to develop this paper into a publishable form. From a purely formal perspective, I applaud the authors for this remarkable work. Moreover, I highly appreciate the meticulous efforts taken by the authors to inform, underpin, and justify the assumptions for their choices of simulation parameters. A wide plethora of sources has been used and overall, I deem this approach as valid and sufficient. Also, the manuscript benefits from a highly understandable language throughout all sections.
This said, I believe there exist several potential areas for improvement and for increasing readability even further. First, the wording between the different players in the platform ecosystem could be clearer. The platform operator (e.g., Amazon) is denoted as “product delivery platform”. Here I would suggest to simply go with “market platform” or “market place”. Calling this entity a (product) DELIVERY platform confuses the reader because Amazon does just NOT deliver the products but draws on complementors (as rightly stated by the authors).
Second, the paper suffers a little bit from the proposed model’s protruding complexity. The authors write several time “for the sake of simplicity” but still, the paper appears to be a bit overly complex. It may be a reasonable idea to examine which of the “modules”, parameters, and elements of the simulation may be disregarded without too much loss.
Third, one major concern is that the manuscript does not do a good job yet in clearly stating (upfront) what the “ecosystem strategy” exactly is. I believe this should be clearly stated in the abstract and in the introduction already. Of course, on page 6 they refer to a definition (Adner [5] (p.47) defines an ecosystem strategy as "the way in which a focal firm approaches the alignment of partners and secures its role in a competitive ecosystem.") but this doesn’t really help as it remains vague. What does concretely mean for the simulated environment? Later on, the authors refine this and state the ecosystem strategy facilitates standardization and cooperation among the physical intermediary firms (i.e., the parcel delivery services), but do not explain precisely what this actually means. Also, within subsection 2.4.1, the refer to subsection 2.4 for an explanation of the ecosystem strategy (which appears odd), but fail to deliver the explanation.
Fourth, Figure 1 and the formulas in the paragraphs leading to Figure 1 (i.e., y=aL-b(x-L), etc.) should be clearly linked. The model parameters should, if possible, be picked up and illustrated within Figure 1. Otherwise, Figure 1 feels disconnected to the text and the reader is completely lost when interpreting what it means.
Author Response
We express our appreciation to the reviewer for such insightful comments, which have helped us significantly improve the paper. We have attempted to address all comments and have revised our manuscript as the attached file.
Changes in the revised manuscript are in red.

Reviewer 2 Report
The article is very interesting and is more than suitable for this journal. However there are some minor topics that the authors should address
Abstract: Please provide some information regarding the theory where the paper adds value to the literature
Introduction
Lines 30-33 etc. Please provide some references
Lines 47-50. How do you feel? Articles should be more rigorous and not based on feelings.
"Traditionally, 49
Japanese logistics firms place much importance on service quality, especiall" ok, but from where do you know that? This is not a general known fact. Please cite!
". Finally, Sagawa stopped its partnership 55
with Amazon in 2013" please cite! This is not general known! It is a very particular information
Please give some proper references for all this information in this paragraph!
" Current research on platforms have also focused on 86
ecosystem perspectives and developed the concept of "platform ecosystem" Which current research? Please cite properly!
The introduction is a little bit to long. The authors need to provide some more information regarding the research gap by highlighting some more references and explaining how the research gap translates into the research question. Please also bring more arguments why you choose to conduct the study in Japan
Lit review
References for figure 1? For figure 3?
My recommendation is that you have a distinct part regarding literature review and another part regarding methodology where you explain what you analysed and how.
Please also provide some references for the methodology that you employ
Discussion
here you should highlight how the paper is different than other references in the literature. Discussion is always a comparison of your results to results obtained by others
Conclusions
You need to have here 4 sections
Implications for theory - how does the paper add value to knowledge / to existing literature Implications for practice Limitations Future research perspectives
You should try to link better the paper to the scope of the journal
So... please restructure the article accordingly and add some conclusions. A paper without conclusions is like a story without and end
Author Response

(The authors gave the same response as above.)

Round 2
Reviewer 2 Report
The manuscript has been considerably improved. Congratulations for the good work.
Author Response
We again wish to express our gratitude for your comments on our paper.
Those were significantly helpful to improve the article.